# SURE: Scalable Uncertainty Estimation for Multimodal Pretrained Pipelines with Missing Inputs

## Abstract

Pretrained multimodal models offer strong representational priors and sample efficiency, but remain fragile when deployed in real-world settings. Two key challenges underlie this brittleness: (1) inputs are frequently incomplete due to missing or corrupted modalities, and (2) pretrained models may yield unreliable predictions due to distribution mismatch or insufficient adaptation. A common workaround for the first challenge is to reconstruct missing modalities; however, this alone not only fails to resolve the second challenge, but may exacerbate it–introducing additional uncertainty from reconstruction that compounds the inherent unreliability of the pretrained model. We propose **SURE** (Scalable Uncertainty and Reconstruction Estimation), a lightweight, plug-and-play module that enhances pretrained multimodal pipelines with deterministic latent-space reconstruction and principled uncertainty estimation. SURE decomposes prediction uncertainty into two sources: *input-induced uncertainty*, traced from reconstruction via error propagation, and *model mismatch uncertainty*, reflecting the limits of the frozen model. To support stable uncertainty learning, SURE employs a distribution-free Pearson correlation-based loss that aligns uncertainty scores with reconstruction and task errors. Evaluated on both a tractable linear-Gaussian toy problem and several real-world tasks, SURE improves prediction accuracy and uncertainty calibration, enabling robust, trust-aware inference under missing or unreliable input conditions.

## 1 Introduction

Multimodal learning has emerged as a powerful paradigm for integrating diverse information sources (Zong & Sun, 2023; Wan et al., 2023), often outperforming unimodal models in expressiveness and generalization (Huang et al., 2021). Yet despite these advances, *pretrained multimodal models remain underutilized in practice*. A key reason lies in their fragility under realistic, resource-constrained deployment conditions. First, these models assume that all modalities are consistently present and of high quality–an assumption rarely satisfied in real-world scenarios such as autonomous driving, healthcare, or wearable computing. In such environments, sensor failures, occlusions, or transmission errors frequently lead to missing or corrupted inputs. Since pretrained multimodal pipelines typically rely on tightly coupled fusion mechanisms, even a single missing modality can trigger cascading failures and significant performance degradation. Second, pretrained models often retain biases or mismatches that exist in their original training data or arise during adaptation to downstream tasks (e.g., due to distribution shift), leading to unreliable predictions even on seemingly complete inputs. This problem is often silently ignored with the lack of calibrated confidence estimates, leaving users unable to assess the reliability of a given prediction.

Despite these limitations, pretrained multimodal models offer strong representational priors and data efficiency, especially in low-resource or high-dimensional regimes. In Section 5.1, we highlight the benefit of utilizing pretrained multimodal models over training from scratch. Therefore, rather than discarding pretrained models, our goal is to *preserve their utility while mitigating their risks* by equipping them with lightweight modules that address two key gaps: (1) incomplete inputs and (2) uncalibrated model confidence.

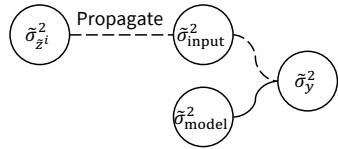

Figure 1: Decomposition of prediction uncertainties.

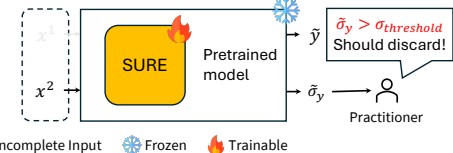

Figure 2: Simple decision making process with SURE's estimated uncertainty.

To address (1), a common strategy to handle missing modalities is to reconstruct them from the available ones, allowing the pretrained pipeline to operate as if the input were complete. Previous efforts in this direction include latent imputation (Lian et al., 2023), prompt-based tuning (Lee et al., 2023), and masked autoencoding (Woo et al., 2023). While effective at syntactically completing the input, these methods do not evaluate the uncertainty of the reconstructed inputs and how these uncertainties would affect the final predictions. This omission introduces a critical failure mode: *the model proceeds with high confidence even when reconstructions are uncertain or erroneous.* Combined with the model's inherent biases, *this can amplify unreliability, effectively worsening challenge (2) rather than alleviating it.*

**Uncertainty estimation is thus the missing piece for reliable multimodal inference–directly addressing challenge (2)**, especially when models operate under missing data and imperfect pretrained backbones. Various methods have been proposed to quantify the uncertainty of predictions, including Bayesian deep learning (Wang et al., 2019b; Kendall & Gal, 2017a) or evidential learning (Sensoy et al., 2018), which rely heavily on the assumption about the output distribution. Several studies have also applied post-hoc techniques, such as Deep Ensembles (Lakshminarayanan et al., 2017), which rely on output variance across multiple forward passes. However, these approaches increase latency and hinder practical implementation. Additionally, while effective in unimodal settings, previous uncertainty estimation techniques are rarely adapted to multimodal pipelines. Critically, they overlook the unreliability introduced by reconstructing missing modalities. In this paper, we provide an ablation study (Section 5.1) to highlight the importance of estimating the uncertainty of the reconstructed inputs.

To jointly address both challenges, we propose **SURE** (Scalable Uncertainty and Reconstruction Estimation), a *compact*, *plug-and-play* module that augments pretrained multimodal models with *reliable missing-modality handling* and *principled uncertainty estimation*. Our framework imposes no constraints or assumptions on the output distribution or backbone architecture, demonstrating its effectiveness across a wide range of multimodal model architectures and downstream tasks (as shown in Section 4). At its core, SURE introduces a deterministic **latent-space reconstruction module** that estimates missing modalities from the available ones. This reconstruction operates entirely in the shared latent space of the pretrained pipeline, minimizing disruption to the frozen architecture and enabling seamless reuse of existing weights.

Crucially, SURE goes beyond reconstruction by estimating **prediction uncertainty**, either with complete or incomplete inputs, which is decomposed into two complementary components mentioned earlier:

- **Input-induced uncertainty** ($\tilde{\sigma}^2_{\text{input}}$): error introduced by uncertain reconstructed inputs;
- **Model mismatch uncertainty** ($\tilde{\sigma}^2_{\text{model}}$): error due to the inherent imperfection of the pretrained model.

*Explicitly modeling these two uncertainty sources brings two key advantages.* First, it enables finer-grained diagnosis of prediction reliability: we can distinguish whether uncertainty stems from noisy inputs or from limitations of the pretrained model itself. Second, it supports more interpretable and actionable decision-making, e.g., deciding whether to seek better input signals or to defer prediction due to model limitations. This decomposition is detailed in Section 2.4, where we formally define the modeling choices that allow SURE to isolate and quantify these factors in a principled way.

To obtain input-induced uncertainty, SURE first estimates **reconstruction uncertainty** ($\tilde{\sigma}^2_{\tilde{z}^i}$) for each missing modality–quantifying the confidence in its latent reconstruction. This uncertainty is then propagated through the frozen pretrained network using an **adaptation of classical error propagation theory**,

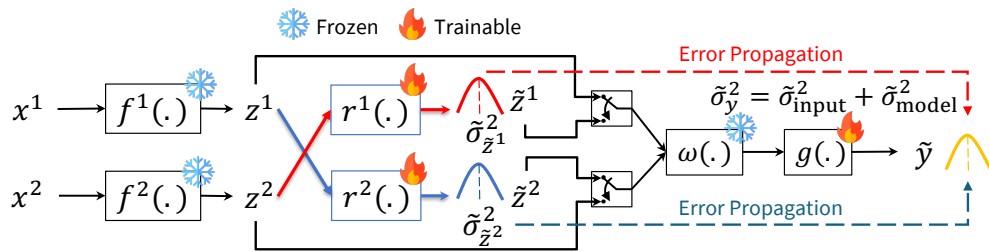

Figure 3: Overview of SURE. Reconstruction modules $r^i(.)$' are inserted after latent projections $f^i(.)$ and before fusion layers $\omega(.)$ in pretrained multimodal frameworks, reconstructing missing modalities with uncertainties. Reconstructed outputs replace missing inputs while propagating uncertainty, and the final classifier estimates output uncertainty from the model's bias. The notation $\widetilde{\bullet}$ distinguishes SURE's estimates from inputs, outputs, and frozen module's intermediate variables.

yielding a principled estimate of $\tilde{\sigma}^2_{\text{input}}$. In parallel, the classifier head is extended to output both the final prediction and a residual **model mismatch uncertainty** ($\tilde{\sigma}^2_{\text{model}}$), reflecting factors unrelated to the input quality. These two sources are then combined to yield the total prediction uncertainty (illustrated in Figure 1), enabling downstream decisions based on model confidence (Figure 2). Finally, to enable stable and distribution-free training of these uncertainty scores, SURE employs a **Pearson correlation-based loss** that aligns predicted uncertainties with actual errors–reconstruction errors for $\tilde{\sigma}^2_{\tilde{z}^i}$ and task errors for $\tilde{\sigma}^2_y$. This sidesteps the limitations of likelihood-based uncertainty estimation while promoting tight correlation between uncertainty and predictive risk.

We evaluate SURE in two stages. First, we validate its uncertainty estimation in a toy linear-Gaussian setup where all quantities admit closed-form solutions. SURE closely matches the oracle variance and corrects for frozen model bias. Then, on three real-world benchmarks–sentiment analysis, book genre classification, and human activity recognition–we demonstrate that SURE consistently improves both predictive performance and uncertainty calibration under various missing-modality scenarios.

**To summarize, our contributions are as follows:**

- We propose SURE, a unified framework that augments pretrained models with latent-space reconstruction and uncertainty estimation to improve robustness under missing modalities.
- We adapt classical error propagation theory and introduce a Pearson correlation-based objective to estimate predictive uncertainty without strong distributional assumptions.
- We demonstrate, both analytically and empirically, that SURE improves decision reliability and supports uncertainty-informed inference across a range of multimodal tasks on a variety of state-of-the-art multimodal models.

## 2 The Proposed Method

### 2.1 Problem Formulation

Let $\Phi(\cdot)$ denote a multimodal pipeline, pretrained on large-scale, complete datasets for a specific supervised task. Our objective is to extend this pipeline to operate reliably in more realistic, imperfect conditions– where some input modalities may be missing. Formally, let the dataset be $\mathcal{D} = \{(\mathbf{x}_i, \mathbf{y}_i)\}_{i=1}^N$ where $N$ is the number of samples in the dataset, each sample $\mathbf{x}_i = \{\mathbf{x}_i^1, \ldots, \mathbf{x}_i^M\}$ consists of $M$ modalities from the domain $\mathcal{X} = \mathbb{R}^{n_1} \times \cdots \times \mathbb{R}^{n_M}$, and the corresponding output label is $\mathbf{y}_i \in \mathcal{Y} = \mathbb{R}^k$. In practical deployments, a subset of modalities may be missing for each input. Let $\mathcal{A}_i \subseteq \{1, \ldots, M\}$ denote the index set of modalities available for sample $i$, and let $\bar{\mathbf{x}}_i = \{\mathbf{x}_i^a\}_{a \in \mathcal{A}_i}$ be the corresponding missing modality input subset. In the rest of the manuscript, index $i$ would be discarded for notational simplicity when discussing one sample. We aim to augment the pretrained model $\Phi(\cdot)$ with the following modifications:

1. **Reconstruction of missing modalities** from the available subset $\bar{\mathbf{x}}$, while preserving the structure and utility of the original model.

2. **Estimation of prediction uncertainty** that reflects the model's confidence in its output given the potentially incomplete input.

## 2.2 Overview Framework

We introduce SURE, a lightweight augmentation to pretrained multimodal pipelines. SURE operates entirely in the latent space and is designed to preserve the structure and predictive utility of the original model. As illustrated in Figure 3 (shown for a two-modality case; generalized to arbitrary $M$ in Appendix A.2.3), SURE adds two key components to the frozen backbone:

- **Reconstruction modules** $r^j(\cdot)$, one for each modality, inserted after the frozen unimodal encoders. When a modality $j$ is missing, $r^j(\cdot)$ estimates its latent representation ($\tilde{\mathbf{z}}^j$) using the available ones and concurrently outputs an uncertainty estimate associated with the reconstruction ($\tilde{\sigma}^2_{\tilde{z}^j}$).
- A modified **classifier head** $g(\cdot)$, which produces both the final prediction ($\tilde{\mathbf{y}}$) and a residual uncertainty estimate ($\tilde{\sigma}^2_{\text{model}}$) that accounts for the imperfect nature of the frozen model.

SURE thus meets the two desiderata outlined in Section 2.1 with minimal modification: the frozen encoders and fusion logic are left intact, while a small set of lightweight modules is introduced for reconstruction and uncertainty estimation.

The following subsections describe SURE's key components in detail: latent-space reconstruction (Section 2.3) and uncertainty estimation logic (Section 2.4), which together enable robust, trust-aware predictions under missing data.

## 2.3 Latent Space Reconstruction Logic

To mitigate performance degradation caused by missing modalities, SURE introduces an efficient and modular reconstruction procedure that operates within the shared latent space of a pretrained multimodal framework. Let $\mathbf{z}^k$ denote the latent representation of the $k$-th modality, produced by the pretrained unimodal projector $f^k(\cdot)$ (as illustrated in Figure 3). In scenarios where $\mathbf{z}^k$ is available, the pretrained pipeline can use it directly in subsequent fusion and classification steps. However, when the $k$-th modality is absent, SURE employs a reconstruction module $r^k(\cdot)$ to estimate $\mathbf{z}^k$ using latent information from an available modality $\mathbf{z}^j$ ($j \neq k$), which yields an approximation $(\tilde{\mathbf{z}}^k; \tilde{\sigma}^2_{\tilde{z}^k}) = r^k(\mathbf{z}^j)$. Here, $\tilde{\mathbf{z}}^k \approx \mathbf{z}^k$ is desirable, and $\tilde{\sigma}^2_{\tilde{z}^k}$ is the uncertainty correspond to this reconstruction, which is later used to quantify input-induced uncertainty (Section 2.4). This reconstruction logic is applied for all $k \in 1, \ldots, M$, and scales linearly with the number of modalities, i.e., $\mathcal{O}(M)$, making it computationally efficient. Detailed architectural designs and complexity analyses of the reconstruction modules are provided in Appendix A.2.2.

Although these modules are architecturally simple, we argue that such a design is well-suited to pretrained multimodal pipelines. Most of these pretrained frameworks (e.g. MLLMs) are deterministic and already learn semantically aligned latent spaces across modalities (Radford et al., 2021; Wu et al., 2024a; Bica et al., 2024). Therefore, when a modality is missing, the reconstruction goal is not to fully re-generate content, but rather to *perturb the latent space minimally and consistently–filling in only the necessary modality-specific information.* This compatibility between deterministic latent reconstruction and the frozen pretrained fusion logic allows SURE to preserve downstream task-relevant signals more effectively than stochastic or overparameterized alternatives, which are often misaligned with fixed backbones. These claims are supported by extensive empirical evidence in Section 5 and Appendix A.4, demonstrating that our lightweight reconstruction design achieves strong alignment and performance under a variety of missing-modality conditions.

## 2.4 Uncertainty Estimation Logic

**Preliminaries.** Uncertainty quantifies the degree of imprecision or doubt associated with an estimate, and is widely studied in engineering analysis and scientific modeling (Aslett et al., 2022; Kumar et al., 2023). Given an incomplete input $\bar{\mathbf{x}} \in \mathcal{D}$, the uncertainty in the resulting prediction $\tilde{\mathbf{y}}$ can be expressed as the variance of the predictive distribution $p(\tilde{\mathbf{y}}|\bar{\mathbf{x}})$.

A conventional approach to uncertainty modeling assumes a fixed form for the predictive distribution $p(\tilde{\mathbf{y}} \mid \overline{\mathbf{x}})$–typically selected for its analytical convenience–and uses the resulting parametric variance as a proxy for prediction uncertainty. The most common choice is a heteroscedastic Gaussian (Upadhyay et al., 2022; Kendall & Gal, 2017c), where the model learns both a mean prediction $\tilde{\mathbf{y}}$ and an associated variance $\tilde{\sigma}_{\tilde{y}}^2$ as parameters of a normal distribution, i.e. $p(\tilde{\mathbf{y}}|\overline{\mathbf{x}}) = \mathcal{N}(\tilde{\mathbf{y}}; \tilde{\sigma}_{\tilde{y}}^2)$. The model parameters $\theta$ are typically learned by maximizing the likelihood over the training dataset:

$$\theta^* = \underset{\theta}{\arg\max} \prod_{(\mathbf{x},\mathbf{y}) \in \mathcal{D}} \frac{1}{\sqrt{2\pi\tilde{\sigma}_{\tilde{y}}^2}} \exp\left(-\frac{||\tilde{\mathbf{y}} - \mathbf{y}||^2}{2\tilde{\sigma}_{\tilde{y}}^2}\right), \tag{1}$$

which is equivalent to minimizing the negative log-likelihood (NLL) loss:

$$\mathcal{L}_{NLL} = \sum_{(\mathbf{x},\mathbf{y}) \in \mathcal{D}} \frac{||\tilde{\mathbf{y}} - \mathbf{y}||^2}{2\tilde{\sigma}_{\tilde{y}}^2} + \frac{\log(\tilde{\sigma}_{\tilde{y}}^2)}{2}. \tag{2}$$

In this formulation, the optimal variance estimate takes a closed-form solution equal to the squared prediction error:

$$\tilde{\sigma}_{\tilde{y}}^{2*} = ||\tilde{\mathbf{y}} - \mathbf{y}||^2 := \Delta_y^2. \tag{3}$$

While widely adopted, this approach presents several important limitations:

- *Restrictive distributional assumptions:* It presumes a specific form (e.g., Gaussian) for $p(\tilde{\mathbf{y}} \mid \tilde{\mathbf{x}})$, which may not hold in practice, leading to poorly calibrated uncertainty estimates. This can be easily observed in the experimental result in Section 4.
- *Numerical instability at low error:* When the prediction error $\Delta_y^2 \to 0$, the gradient of $\mathcal{L}_{NLL}$ becomes ill-defined (i.e the gradient has the form $\frac{0}{0}$), making uncertainty learning unstable (see Appendix A.2.1).
- *Magnitude entanglement:* The predicted uncertainty $\tilde{\sigma}_{\tilde{y}}^2$ is forced to match the squared error $\Delta_y^2$ exactly–an ideal but overly rigid constraint that is often difficult to satisfy during optimization (see Section 5.2).

**Distribution-free Uncertainty Estimation.** Unlike conventional likelihood-based methods, which estimate prediction uncertainty directly by assuming a specific output distribution, our approach models the total uncertainty $\tilde{\sigma}_{\tilde{y}}^2$ into two interpretable components:

$$\tilde{\sigma}_{\tilde{y}}^2 = \tilde{\sigma}_{\text{input}}^2 + \tilde{\sigma}_{\text{model}}^2, \tag{4}$$

where:

- The *input-induced uncertainty* $\tilde{\sigma}_{\text{input}}^2$ is analytically derived by propagating reconstruction uncertainty from missing latent variables $(\tilde{\sigma}_{\tilde{z}j}^2)$.
- The *model mismatch uncertainty* $\tilde{\sigma}_{\text{model}}^2$ is directly estimated by SURE's modified prediction head, capturing residual variance due to biases from the pretrained model itself.

Here, we make a deliberate modeling choice: we treat the uncertainty inherent to fully observed, clean inputs as *irreducible noise*–a fixed lower bound that is not modeled explicitly. Instead, our focus is on the additional uncertainty introduced by reconstruction and pretrained model mismatch. This allows us to isolate the components of uncertainty that are actionable and relevant in deployment settings involving missing or corrupted inputs.

Importantly, our estimated uncertainty $\tilde{\sigma}_{\tilde{y}}^2$ is not intended to recover the absolute point-wise variance of the full predictive distribution. Rather, it serves as a *relative confidence indicator*: between any two predictions, a higher uncertainty score corresponds to a higher expected error. This interpretation aligns well with downstream uses such as decision deferral, risk-aware inference, or selective prediction, where relative ranking of confidence is more valuable than calibrated absolute variance.

**Input-Induced Uncertainty via Error Propagation.** When input modalities are missing, the reconstruction process introduces uncertainty into the model pipeline. Drawing from the classical Error Propagation formulation (Arras, 1998; Tellinghuisen, 2001), we establish the following result:

**Proposition 2.1.** *Let $\mathcal{U}$ denote the sets of indices corresponding to missing modalities and $\tilde{\mathbf{z}}^j (j \in \mathcal{U})$ be the reconstructed latent representations for missing modalities, each with associated reconstruction uncertainty $\tilde{\sigma}^2_{\tilde{z}^j}$. Assume (i) the reconstruction error $\Delta\mathbf{z}^j = \mathbf{z}^j - \tilde{\mathbf{z}}^j$ is zero-mean with covariance $\mathrm{Var}(\Delta\mathbf{z}^j) = \tilde{\sigma}^2_{\tilde{z}^j}\mathbf{I}$, (ii) the magnitudes of errors $\|\Delta\mathbf{z}^j\|$ are sufficiently small, and (iii) errors across modalities are independent. Let $\Omega(\cdot)$ represent the post-reconstruction portion of the pretrained + SURE pipeline (e.g., fusion and task-specific layers), which consumes both available and reconstructed latents. Then, under a first-order approximation, the total input-induced uncertainty in the output is:*

$$\tilde{\sigma}^2_{input} \approx \sum_{j \in \mathcal{U}} \|\nabla_{\tilde{\mathbf{z}}^j}\Omega\|^2 \tilde{\sigma}^2_{\tilde{z}^j}. \tag{5}$$

***Sketch-Proof.***

Applying a first-order Taylor expansion:

$$\Omega(\{\mathbf{z}^i\}, \{\tilde{\mathbf{z}}^j + \Delta\mathbf{z}^j\}) = \Omega(\mathbf{z}) + \sum_{j \in \mathcal{U}} (\nabla_{\tilde{\mathbf{z}}^j}\Omega)^\top \Delta\mathbf{z}^j + O(\|\Delta\mathbf{z}^j\|^2).$$

Since $\mathbf{z}^i$ (for $i \in \mathcal{A}$) are fixed, only the linear term contributes to the variance:

$$\tilde{\sigma}^2_{\text{input}} \approx \mathrm{Var}\Big[\sum_{j \in \mathcal{U}} (\nabla_{\tilde{\mathbf{z}}^j}\Omega)^\top \Delta\mathbf{z}^j\Big] \approx \sum_{j \in \mathcal{U}} \|\nabla_{\tilde{\mathbf{z}}^j}\Omega\|^2 \tilde{\sigma}^2_{\tilde{z}^j},$$

where cross terms vanish by independence. Second-order terms are $O(\sigma^2)$ smaller and are omitted. □

This result enables us to analytically estimate the input-induced uncertainty $\tilde{\sigma}^2_{\text{input}}$ directly from the reconstruction uncertainties $\tilde{\sigma}^2_{\tilde{z}^j}$. Combined with the model mismatch uncertainty $\tilde{\sigma}^2_{\text{model}}$ returned by SURE's prediction head, we obtain the final prediction uncertainty $\tilde{\sigma}^2_{\tilde{y}}$.

**Distribution-free Pearson correlation-based loss.** To learn both reconstruction and prediction uncertainties in a unified manner, we introduce a distribution-free loss based on the *Pearson Correlation Coefficient* (**PCC**). This loss enforces strong correlation between predicted uncertainty $\tilde{\sigma}^2$ and the observed squared error $\Delta^2$, without imposing strict magnitude constraints.

The loss is defined as:
$$\mathcal{L}_{PCC}(\tilde{\sigma}^2, \Delta^2) = 1 - r(\tilde{\sigma}^2, \Delta^2), \tag{6}$$

$$r(\tilde{\sigma}^2, \Delta^2) = \frac{\sum_{i=1}^N (\tilde{\sigma}_i^2 - \mu_{\sigma^2})(\Delta_i^2 - \mu_{\Delta^2})}{\sqrt{\sum_{i=1}^N (\tilde{\sigma}_i^2 - \mu_{\sigma^2})^2} \cdot \sqrt{\sum_{i=1}^N (\Delta_i^2 - \mu_{\Delta^2})^2}}. \tag{7}$$

Here, $\mu_{\sigma^2}$ and $\mu_{\Delta^2}$ denote the batch-wise means of predicted uncertainty and squared error, respectively.

**Proposition 2.2.** *Let $\bar{\sigma}_i^2$ and $\bar{\Delta}_i^2$ be the standardized uncertainty and squared error within a mini-batch: $\bar{\sigma}_i^2 = \frac{\tilde{\sigma}_i^2 - \mu_\sigma}{\sqrt{\frac{1}{N-1}\sum_{j=1}^N (\tilde{\sigma}_j^2 - \mu_{\sigma^2})^2}}, \bar{\Delta}_i^2 = \frac{\tilde{\Delta}_i^2 - \mu_{\epsilon^2}}{\sqrt{\frac{1}{N-1}\sum_{j=1}^N (\bar{\Delta}_j^2 - \mu_{\epsilon^2})^2}}$. Then, $\mathcal{L}_{PCC}$ is approximately equivalent to the MSE between standardized uncertainty and error:*

$$\frac{1}{2N}\sum_{i=1}^N (\bar{\sigma}_i^2 - \bar{\Delta}_i^2)^2 \approx \mathcal{L}_{PCC}(\tilde{\sigma}^2, \tilde{\Delta}^2). \tag{8}$$

***Sketch-Proof.*** Expanding the LHS:

$$\frac{1}{2N}\sum_{i=1}^N \big(\bar{\sigma}_i^2 - \bar{\Delta}_i^2\big)^2 = \frac{1}{2N}\left((2N-2) - 2\sum_{i=1}^N \bar{\sigma}_i^2 \bar{\Delta}_i^2\right) = \frac{2N-2}{2N}(1 - r(\tilde{\sigma}^2, \tilde{\Delta}^2)) = \frac{2N-2}{2N}\mathcal{L}_{PCC}.$$

□

Proposition 2.2 suggests our loss's relaxed constraint allows uncertainty to function as a meaningful confidence indicator–higher predicted uncertainty typically corresponds to larger errors, while lower uncertainty signals more reliable predictions. Furthermore, as shown in Appendix A.2.1, our loss mitigates the gradient instability and miscalibration issues that affect standard Gaussian NLL objectives near optimality, promoting more stable learning even in low-error regimes.

SURE applies the same $\mathcal{L}_{\text{PCC}}$ objective to both reconstruction and prediction uncertainty, promoting a strong correlation between predicted uncertainty and observed error in each case:

- For *reconstruction uncertainty*, we apply:

$$\mathcal{L}_{\text{rec}}^{j} = \frac{1}{N} \sum |\widetilde{\mathbf{z}}^{j} - \mathbf{z}^{j}|^2 + \lambda \cdot \mathcal{L}_{PCC}(\tilde{\sigma}_{\tilde{z}^j}^2, \Delta_{\tilde{z}^j}^2), \tag{9}$$

  where $\Delta_{\tilde{z}^j}^2 = |\widetilde{\mathbf{z}}^{j} - \mathbf{z}^{j}|^2$ denotes the reconstruction error, and $\lambda$ controls the uncertainty supervision strength.
- For *prediction uncertainty*, a similar objective is used:

$$\mathcal{L}_{\text{task}} = \mathcal{L}_{downstream}(\tilde{\mathbf{y}}, \mathbf{y}) + \lambda \cdot \mathcal{L}_{PCC}\left(\tilde{\sigma}_{\tilde{y}}^2, \Delta_y^2\right), \tag{10}$$

  where $\Delta_y^2$ is the task-specific prediction error–e.g., mean squared error for regression, or cross-entropy loss for classification.

**Training process.** In the first phase, reconstruction modules are trained with $\mathcal{L}_{rec}$ using one modality as ground truth and others as input. In the second phase, reconstruction modules are frozen, and the classifier head is trained with $\mathcal{L}_{\text{task}}$. Detailed training steps are in Algorithm 1 (Appendix A.2.3).

## 3 Analytic sanity check in a linear–Gaussian toy world

Before tackling realistic tasks, we first design a *linear–Gaussian* scenario in which *every quantity of interest— reconstruction variance, propagated input variance, oracle output variance— admits tractable closed-form expressions*. This analytically tractable sandbox lets us verify that SURE (1) could reproduce the oracle distributional variance and even capture fine-grain instance-level prediction variance; (2) improves the frozen model's prediction error.

### 3.1 Experiment setup

**Synthetic data generation.** Table 1 specifies a simple two-dimensional *linear–Gaussian world*. The latent fusion signal is $\omega = w_1 x_1 + w_2 x_2$ and serves directly as the regression target $y = \omega$. We draw 10000 i.i.d. samples and mask modality $x_2$ in 50% of them; 80% of the samples are used to learn SURE's auxiliary modules and the remaining 20% are held out for testing.

<table>
<tr><td colspan="3">Table 1: Gaussian toy data parameters.</td><td colspan="3">Table 2: Frozen model's parameters.</td></tr>
<tr><td>Symbol</td><td>Definition</td><td>Value</td><td>Symbol</td><td>Definition</td><td>Value</td></tr>
<tr><td>$x_1$</td><td>$\mathcal{N}(0,1)$</td><td>—</td><td>$\tilde{\omega}$</td><td>$\tilde{w}_1 x_1 + \tilde{w}_2 x_2$</td><td>$\tilde{w}_1 = w_1 + \delta_1$</td></tr>
<tr><td>$x_2$</td><td>$\rho\, x_1 + z, \; z \sim \mathcal{N}(0, \sigma_z^2)$</td><td>$\rho = 0.8, \; \sigma_z^2 = 0.36$</td><td></td><td></td><td>$\tilde{w}_2 = w_2 + \delta_2$</td></tr>
<tr><td>$\omega$</td><td>$w_1 x_1 + w_2 x_2$</td><td>$w_1 = 2, \; w_2 = 3$</td><td></td><td></td><td>$(\delta_1, \delta_2) = (0.5, -0.7)$</td></tr>
<tr><td>$y$</td><td>$\omega$</td><td>—</td><td>$\tilde{y}$</td><td>$\tilde{\omega}$</td><td>–</td></tr>
</table>

**Frozen biased model.** To replicate our motivation setting in which SURE would be incorporated with a frozen model, a biased model is adopted instead of the ideal fusion logic we use to synthesize the data (Table 2). This pipeline consists of a frozen linear fusion $\tilde{\omega} = \tilde{w}_1 x_1 + \tilde{w}_2 x_2$; when $x_2$ is missing, whatever value is supplied by an external reconstructor is used; and an identity prediction head $\tilde{y} = \tilde{\omega}$.

**The bias introduced by the frozen model.** As we assume the imperfection of the frozen model, there exists irreducible model-mismatch variance. With the weight error $\delta = (\delta_1, \delta_2)$, for arbitrary $(x_1, x_2)$ input, the frozen prediction $\tilde{y} = \omega + \delta_1 x_1 + \delta_2 x_2$ exhibits *model-mismatch variance*

$$\sigma_{\text{model}}^2 = \delta^\top \text{Cov}\big[(x_1, x_2)^\top\big]\delta = 0.18, \tag{11}$$

even when both modalities are present. This irreducible error term motivates the need for a new *prediction head* with uncertainty estimation, which SURE provides.

**Unbiased reconstruction and error propagation.** With the true knowledge of the data generation model, we could build an optimal unbiased reconstructor for $x_2$ when this modality is missing. In this setup, this reconstructor is the MMSE estimator $\tilde{x}_2 = \mathbb{E}[x_2 \,|\, x_1] = \rho x_1$ with residual variance $\sigma_{x_2}^2 = \text{Var}(x_2 \,|\, x_1) = \sigma_z^2$. Propagating this through the frozen predictor quantifies the uncertainty of the prediction, caused by the reconstructed data:

$$\sigma_{\text{input}}^2 = \left(\frac{\partial \tilde{y}}{\partial \tilde{x}_2}\right)^2 \sigma_{x_2}^2 = \tilde{w}_2^2 \sigma_z^2 = 1.90. \tag{12}$$

In total, if we adopt the ideal MMSE estimator as the reconstructor, and fully leverage the frozen model as the predictor, we could get the oracle total variance in closed-form as follow:

$$\sigma_{y,\text{oracle}}^2 = \sigma_{\text{model}}^2 + \sigma_{\text{input}}^2 \mathbf{1}_{\{x_2 \text{ reconstructed}\}} = \begin{cases} 0.18, & x_2 \text{ observed,} \\ 0.18 + 1.90 = 2.08, & x_2 \text{ reconstructed.} \end{cases} \tag{13}$$

## 3.2 Methods compared

With this setup, we focus on comparing two models:

**Frozen model + Oracle closed-form solution of variance.** This baseline uses the optimal reconstructor and the frozen model as the predictor and uses Equation 13 to produce variance — this represents an unattainable upper bound on uncertainty quality.

**SURE (ours).** A linear reconstructor $r_2(x_1) \to (\tilde{x}_2, \tilde{\sigma}_{x_2}^2)$ is adopted, replacing the optimal MMSE. In parallel, a linear prediction head $g(x_1, x_2) \to (\tilde{y}_{SURE}, \tilde{\sigma}_{\text{model}}^2)$ replaces the original prediction head $\tilde{y} = \tilde{\omega}$ of the frozen model. Both of these layers are trained with the PCC losses and MSE loss. For SURE, the total deployed variance is $\tilde{\sigma}_y^2 = \tilde{\sigma}_{\text{model}}^2 + \tilde{w}_2^2 \tilde{\sigma}_{x_2}^2 \mathbf{1}_{\{x_2 \text{ missing}\}}$.

Through this comparison, we could answer the following questions:

1. **Downstream task performance:** Checks whether the auxiliary heads that SURE adds *improve or at least preserve* the task loss. A lower MSE shows that re-estimating the bias term $g(\tilde{\omega})$ corrects the systematic error of the frozen model rather than harming it.

2. **Uncertainty fidelity:** Asks whether SURE (i) *reproduces the population-level* variance that could be calculated with perfect knowledge of the data-generating process and the frozen weights, and (ii) *sharpens the sample-level* variance assigned to each individual prediction.

3. **Error correspondence:** Verifies that higher predicted variance genuinely signals larger errors on a sample-level basis. A positive PCC close to 1 means the model's confidence ranking aligns with actual mistakes.

## 3.3 Empirical verification

On the 20% held-out split we quantify both *task accuracy* and *uncertainty quality*. Specifically, we report (i) **Mean-Squared Error (MSE)** of the point predictions; and (ii) **Uncertainty–Calibration Error (UCE)** and the **Pearson correlation (PCC)** between predicted variance and squared error, which together reflect how well the variance tracks the realized error.

Table 3: SURE versus Frozen model + Oracle closed-form variance with toy dataset.

| Method | MSE↓ | UCE↓ | PCC(err$^2$, var)↑ |
|---|---|---|---|
| Frozen + Oracle | 1.7323 | 1.5523 | 0.4316 |
| SURE | **1.7193** | **0.6370** | **0.4335** |

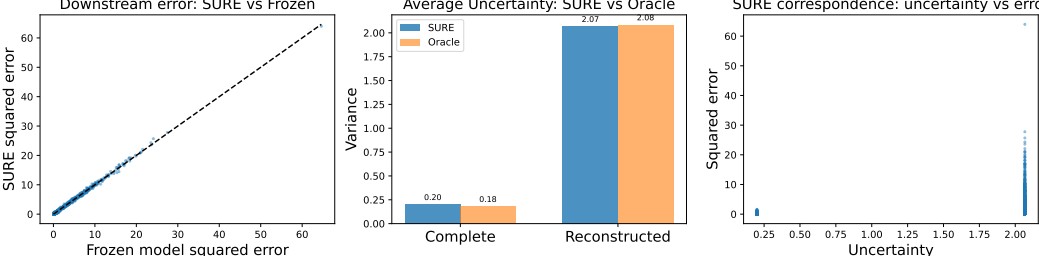

Figure 4: SURE performance on toy dataset.

**Results.** Table 3 summarizes the numbers and Figure 4 illustrates these comparisons visually:

In short, the toy experiment demonstrates that SURE can *simultaneously* refine point predictions *and* provide substantially better-calibrated uncertainty, matching the analytic oracle within sampling noise:

*(i) Downstream accuracy.* Replacing the frozen head by SURE's trainable head yields a small but consistent MSE reduction. Although the toy setting is noise-free, this shows that learning to compensate for the frozen weights does not harm and slightly improves point accuracy.

*(ii) Uncertainty fidelity.* UCE decreases from 1.55 to 0.64, a 59% **improvement**, indicating that SURE is much better calibrated to the oracle uncertainty on a *per-sample* basis. When averaged across the test set (Fig. 4), the predicted and oracle variance bars are virtually indistinguishable, demonstrating that SURE recovers the oracle distribution almost exactly. In addition, the positive correlation (PCC = 0.43) shows that samples assigned higher uncertainty also incur larger squared error, thus meeting the error–correspondence test.

## 4 Evaluation on nonlinear pretrained models and real benchmarks

### 4.1 Experiment settings

To demonstrate the scalability and distribution-free uncertainty estimation capability of SURE under practical settings, we integrate it into three pretrained multimodal frameworks and adapt them to smaller-scale datasets with missing modalities during training and evaluation. Detailed integration settings are provided in Appendix A.2.4.

**Sentiment Analysis.** For this task, we use MMML (Wu et al., 2024b), a state-of-the-art architecture pretrained on CMU-MOSEI (Zadeh et al., 2018), and fine-tune it on CMU-MOSI (Zadeh et al., 2016) with missing modalities. Metrics include mean absolute error (MAE), correlation between predicted semantic level and ground truth (Corr), binary accuracy (Acc-2), and F1 score.

**Book Genre Classification.** SURE is coupled with MMBT (Kiela et al., 2019), pretrained on MM-IMDB (Arevalo et al., 2020), and fine-tune it for book genre classification with text and image data from (Haque et al., 2022). Evaluation metrics are accuracy and F1 score.

**Human Action Recognition.** We use the HAMLET framework (Islam & Iqbal, 2020), pretrained on MMAct (Kong et al., 2019), and fine-tune it on UTD-MHAD (Chen et al., 2015) with missing modalities. Metrics include accuracy and F1 score.

The quality of the output Uncertainty estimation task is evaluated using Uncertainty Calibration Error (Output UCE) (Guo et al., 2017) and Pearson Correlation Coefficient (Output Corr) (Upadhyay et al., 2022), which measure alignment between predictive error and uncertainty. We also measure the uncertainty quality for reconstruction modality using the Pearson Correlation Coefficient (R-Corr)

For the main comparisons, we include both reconstruction-based methods–ActionMAE (Woo et al., 2023), DiCMoR (Wang et al., 2023), and IMDer (Wang et al., 2024)–and uncertainty estimation techniques, including Gaussian Maximum Likelihood (Kendall & Gal, 2017b; Wang et al., 2019a), Monte Carlo Dropout (Maddox et al., 2019; Laves et al., 2019; Srivastava et al., 2014), and Deep Ensembles (Lakshminarayanan et al., 2017). To ensure a fair and focused evaluation of uncertainty modeling and reconstruction under pretrained settings, *all baseline methods are integrated into the same pretrained architectures used by SURE.* For Gaussian Maximum Likelihood, we utilize SURE's backbone and replace the correlation-based uncertainty loss $\mathcal{L}_{rec}$ with $\mathcal{L}_{NLL}$. During training, 50% of the samples in each modality are randomly masked, with independent masking patterns applied across modalities.

This standardized setup eliminates confounding factors related to architectural differences or diverging training protocols. While each method's core logic is preserved from its original implementation, performance divergences from their published results arise solely due to our unified masking and evaluation protocol, which isolates the contribution of uncertainty and reconstruction strategies under a consistent pretrained backbone.

Additional baselines and results are discussed in Appendix A.4.

Table 4: Results on Sentiment Analysis task on CMU-MOSI Dataset.

| Model | MAE ↓ | | | Corr ↑ | | | F1 ↑ | | | Acc-2 ↑ | | |
|---|---|---|---|---|---|---|---|---|---|---|---|---|
| | T(ext) | A(udio) | F(ull) | T | A | F | T | A | F | T | A | F |
| ActionMAE | 1.106 | 2.146 | 1.005 | 0.506 | 0.155 | 0.517 | 0.717 | 0.57 | 0.719 | 0.724 | 0.423 | 0.725 |
| DiCMoR | 0.811 | 1.227 | 1.106 | 0.783 | 0.427 | 0.537 | 0.854 | 0.57 | 0.65 | 0.856 | 0.585 | 0.654 |
| IMDer | 0.707 | 1.237 | 1.106 | 0.797 | 0.438 | 0.544 | 0.846 | 0.524 | 0.62 | 0.846 | 0.564 | 0.634 |
| **SURE** | **0.602** | **1.148** | **0.583** | **0.865** | **0.557** | **0.869** | **0.896** | **0.685** | **0.891** | **0.894** | **0.684** | **0.89** |

Table 5: Results on Uncertainty Estimation task on CMU-MOSI Dataset.

| Method | R-Corr ↑ | | Output Corr ↑ | | | Output UCE ↓ | | |
|---|---|---|---|---|---|---|---|---|
| | T | A | T | A | F | T | A | F |
| Gaussian MLE | 0.103 | 0.013 | 0.067 | 0.032 | 0.059 | 0.425 | 0.476 | 0.385 |
| MC DropOut | 0.047 | 0.008 | 0.013 | 0.009 | 0.13 | 0.496 | 0.51 | 0.396 |
| DeepEnsemble | 0.062 | 0.031 | 0.024 | 0.074 | 0.082 | 0.497 | 0.492 | 0.389 |
| **SURE** | **0.739** | **0.732** | **0.381** | **0.18** | **0.485** | **0.315** | **0.429** | **0.285** |

Table 6: Results on Book Gerne Classification task on Book Dataset.

| Model | F1 ↑ | | | Acc ↑ | | |
|---|---|---|---|---|---|---|
| | T(ext) | I(mage) | F(ull) | T | I | F |
| ActionMAE | 0.277 | 0.271 | 0.35 | 0.186 | 0.166 | 0.311 |
| DiCMoR | 0.202 | **0.465** | 0.467 | 0.152 | **0.452** | 0.454 |
| IMDer | 0.204 | 0.376 | 0.374 | 0.155 | 0.368 | 0.367 |
| **SURE** | **0.683** | 0.413 | **0.696** | **0.671** | 0.401 | **0.688** |

Table 7: Results on Uncertainty Estimation task on Book Dataset.

| Method | R-Corr ↑ | | Output Corr ↑ | | | Output UCE ↓ | | |
|---|---|---|---|---|---|---|---|---|
| | T | I | T | I | F | T | I | F |
| Gaussian MLE | 0.137 | 0.233 | 0.358 | 0.349 | 0.468 | **0.193** | 0.198 | 0.115 |
| MC DropOut | 0.243 | 0.334 | 0.174 | 0.186 | 0.41 | 0.249 | 0.222 | 0.134 |
| DeepEnsemble | 0.128 | 0.135 | 0.144 | 0.214 | 0.227 | 0.242 | 0.231 | 0.177 |
| **SURE** | **0.637** | **0.833** | **0.373** | **0.481** | **0.474** | 0.211 | **0.19** | **0.103** |

## 4.2 Main Results

Results show pipeline performance with unimodal or full inputs, averaged over three runs. Best and second-best metrics are highlighted in **red** and blue, respectively.

**Sentiment Analysis.** In the results for the CMU-MOSI dataset, SURE consistently outperforms recent reconstruction techniques (Table 4) and uncertainty estimation methods (Table 5), highlighting their effectiveness in handling missing modalities and estimating the uncertainty of the model. Among the modalities, audio appears to be less effective for the downstream task. All methods perform better when text is available compared to when only audio is used, and uncertainty estimation also declines when relying solely on audio.

**Book Genre Classification.** Similar to the sentiment analysis task, SURE outperforms recent reconstruction techniques in this classification task (Table 6), showing a stronger correlation between uncertainty and error for both reconstruction and downstream tasks (Table 7). In the Book Dataset, the text modality

proves to be highly effective for the downstream task, but it contributes less to uncertainty estimation for both reconstruction and downstream tasks.

**Human Action Recognition.** As suggested in Table 8 and Table 9, SURE consistently delivers the best performance on the downstream task and the uncertainty estimation task across almost all scenarios. Output uncertainty most closely reflects actual error when the Watch Accel modality is available. However, we observe that a modality effective for downstream task performance may not always contribute equally to uncertainty estimation. This is likely due to the independent nature of error distributions across different modality combinations, which leads to a divergence between downstream task performance and uncertainty estimation. An extended report with every input modalities combination is presented in Appendix A.4.1.

Table 8: Results on Human Action Recognition task on UTD-MHAD Dataset.

| Model | F1↑ | | | | Acc ↑ | | | |
|---|---|---|---|---|---|---|---|---|
| | V(ideo) | A(ccel) | G(yro) | F(ull) | V | A | G | F |
| ActionMAE | 0.044 | 0.204 | 0.303 | 0.531 | 0.059 | 0.231 | 0.311 | 0.537 |
| DiCMoR | 0.069 | **0.473** | 0.52 | 0.653 | 0.033 | 0.408 | 0.472 | 0.636 |
| IMDer | 0.089 | 0.157 | 0.141 | 0.687 | 0.069 | 0.158 | 0.145 | 0.689 |
| **SURE** | **0.161** | 0.462 | **0.607** | **0.739** | **0.121** | **0.431** | **0.59** | **0.74** |

Table 9: Results on Uncertainty Estimation task on UTD-MHAD Dataset.

| Method | R-Corr↑ | | | Output Corr↑ | | | | Output UCE↓ | | | |
|---|---|---|---|---|---|---|---|---|---|---|---|
| | V | A | G | V | A | G | F | V | A | G | F |
| Gaussian MLE | 0.166 | 0.115 | 0.056 | 0.122 | 0.476 | 0.147 | 0.292 | 0.451 | 0.233 | 0.351 | 0.281 |
| MC DropOut | 0.122 | 0.135 | 0.171 | 0.136 | 0.486 | 0.223 | 0.512 | **0.274** | 0.149 | 0.257 | 0.137 |
| DeepEnsemble | 0.249 | 0.175 | 0.122 | 0.126 | 0.421 | **0.436** | 0.481 | 0.311 | 0.208 | **0.187** | 0.133 |
| **SURE** | **0.878** | **0.837** | **0.863** | **0.226** | **0.53** | 0.306 | **0.568** | 0.301 | **0.104** | 0.226 | **0.009** |

**Summary.** Across all three benchmarks–sentiment analysis, book genre classification, and human action recognition–SURE consistently outperforms recent reconstruction and uncertainty estimation baselines. Its ability to leverage pretrained pipelines while minimally perturbing the pretrained latent space and dynamically reconstructing missing modalities allows for full data utilization during training and robust performance during inference. Moreover, SURE provides well-aligned uncertainty estimates that reliably correlate with prediction errors, enabling trust-aware decision-making. Additionally, the Gaussian MLE method yields poor performance in several cases, highlighting the advantage of our distribution-free training approach. Notably, our results also reveal that the informativeness of a modality for the downstream task does not always translate into equally useful uncertainty estimates, emphasizing the importance of modeling uncertainty explicitly rather than relying on task performance as a proxy. These findings underscore SURE's strength as a unified, lightweight, and generalizable solution for multimodal reasoning under missing and uncertain inputs.

## 5 Analyses

In this section, we demonstrate some key analyses of SURE and pretrained model pipelines, while additional experiments are covered in Appendix A.4.

### 5.1 Ablation Study.

**Settings.** We analyze the impact of various modules on SURE's performance in both uncertainty estimation and downstream tasks. This analysis includes testing several ablated versions of SURE:

(1) **Modality Reconstruction ablation study:**

    (1a) **Remove $r^i(.)$ modules**: Ignore incomplete samples during training.

    (1b) **Rule-based imputation**: Replace missing modalities with zeros.

(2) **Uncertainty Estimation ablation study:**

    (2a) **Remove uncertainty estimation**: Train $r^i(.)$ with MSE only, no uncertainty estimation.

    (2b) **Remove reconstruction uncertainty**: Train $r^i(.)$ with MSE; omit error propagation logic.

(3) **Pretrained weights ablation study**: Reinitialize and train backbone frameworks from scratch.

**Results:** We present the performance of all SURE variations on the UTD-MHAD dataset in Table 10. Overall, each ablation negatively impacts SURE's performance in its respective tasks. Specifically, ignoring missing modalities (1a) or using simple rule-based imputation (1b) significantly reduces downstream task

Table 10: Ablation study of SURE variations on UTD-MHAD Dataset.

| Model | F1 ↑ | | | | Acc ↑ | | | | R-Corr ↑ | | | O-Corr ↑ | | | |
|---|---|---|---|---|---|---|---|---|---|---|---|---|---|---|---|
| | V | A | G | F | V | A | G | F | V | A | G | V | A | G | F |
| (1a) | - | - | - | 0.151 | - | - | - | 0.098 | - | - | - | - | - | - | 0.124 |
| (1b) | 0.095 | 0.059 | 0.408 | 0.519 | 0.094 | 0.081 | 0.413 | 0.525 | - | - | - | 0.128 | 0.322 | 0.122 | 0.524 |
| (2a) | **0.173** | **0.479** | 0.589 | 0.736 | 0.117 | 0.427 | 0.571 | 0.727 | - | - | - | - | - | - | - |
| (2b) | 0.150 | 0.456 | 0.512 | 0.637 | 0.113 | **0.489** | 0.462 | 0.593 | - | - | - | 0.159 | 0.489 | 0.237 | 0.511 |
| (3) | 0.031 | 0.226 | 0.434 | 0.615 | 0.005 | 0.237 | 0.418 | 0.618 | 0.684 | 0.675 | 0.680 | 0.026 | 0.441 | **0.463** | 0.472 |
| **SURE** | 0.161 | 0.462 | **0.607** | **0.739** | **0.121** | 0.431 | **0.590** | **0.740** | **0.878** | **0.837** | **0.863** | **0.226** | **0.530** | 0.306 | **0.568** |

performance, as incomplete yet labeled data remains underutilized and pretrained frameworks are not effectively leveraged. Additionally, while removing the uncertainty estimation logic has a negligible effect on the final task result (2a), this causes an inability to quantify output uncertainty effectively, ruining our initial effort in producing reliable predictions. Moreover, overlooking the uncertainty of the reconstructed inputs by removing reconstruction uncertainty (2b) leads to a significant drop in performance on the output uncertainty estimation task. This highlights our advantages over the previous approach to uncertainty estimation methods for multimodal frameworks. Lastly, the results from variation (3) indicate that training from scratch can significantly degrade prediction performance, particularly in scenarios with missing modalities. In some cases, simply replacing missing modalities with zeros yields comparable or even superior results to training from scratch with reconstruction modules. This reinforces our motivation: utilizing pretrained weights is more efficient and beneficial, especially for smaller datasets involving similar tasks.

### 5.2 Analyses for estimated uncertainty.

**Convergence Analysis.** We visualize the correlation between estimated uncertainties and prediction errors across all training epochs in Figure 5. Compared to the Negative Log-Likelihood Loss (NLL), $\mathcal{L}_{PCC}$ demonstrates superior performance in both convergence speed and final estimation accuracy. Additionally, the shape of the NLL curve suggests instability, as the correlation trend declines after reaching its peak. Although there are fluctuations, our loss maintains an overall upward trend, eventually stabilizing in the final epochs. These experimental results are highly in accordance with our theoretical analysis of convergence points for NLL loss and our proposed loss (Appendix A.2.1).

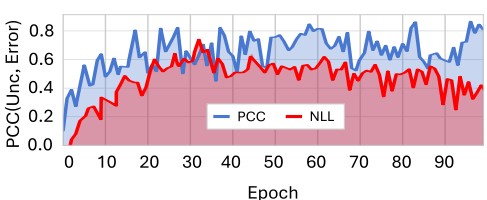

Figure 5: Correlation of estimated uncertainty with prediction error on UTD-MHAD dataset.

**Reconstruction and Output Uncertainty Analysis.** To better trace the major source of prediction error (i.e. either from reconstruction uncertainty or from the model uncertainty), we visualize three quantities across all test samples in the UTD-MHAD dataset, with different modalities combinations where each modality is missing. Ideally, the points should cluster along the bottom-left to top-right diagonal, indicating perfect correlation. With SURE, we observe high efficiency in estimating uncertainty for samples with large prediction errors, which aligns with its intended use as an indicator for potentially error-prone predictions (Figure 6). Notably, when output uncertainties are high, reconstruction uncertainties tend to be elevated as well (Figure 7), suggesting that uncertainties arising from the reconstruction process play a significant role in the overall uncertainty estimation. However, the visualization also indicates a tendency toward overestimating both reconstruction and output uncertainties, highlighting an area for potential improvement in future research.

## 6 Application: Uncertainty-informed Decision Making with SURE

**Settings.** To demonstrate the impact of SURE's uncertainty quantification on decision-making, we simulate this process using a human action recognition task with the UTD-MHAD dataset. SURE is trained with

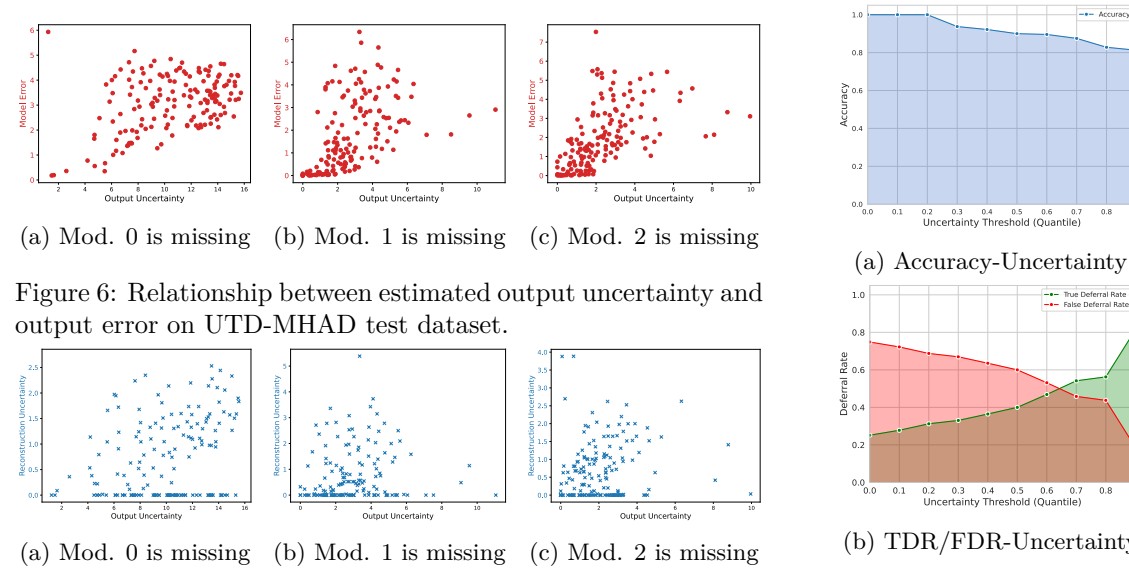

(a) Mod. 0 is missing (b) Mod. 1 is missing (c) Mod. 2 is missing

Figure 6: Relationship between estimated output uncertainty and output error on UTD-MHAD test dataset.

(a) Accuracy-Uncertainty

(a) Mod. 0 is missing (b) Mod. 1 is missing (c) Mod. 2 is missing

Figure 7: Relationship between estimated output uncertainty and reconstruction uncertainty on UTD-MHAD test dataset.

(b) TDR/FDR-Uncertainty

Figure 8: Decision Making Process with Uncertainty on UTD-MHAD Dataset.

similar settings to those used in the main experiment (Table 8). After training, we use the uncertainty estimates to determine whether the model is confident enough to make a final decision or if it should defer the decision for manual inspection. Different uncertainty thresholds are set based on output uncertainty values from the test dataset. For each threshold, predictions with uncertainty higher than the threshold are deferred, and we record **Accuracy**, **True Deferral Rate**, and **False Deferral Rate** (representing the rate of correctly and incorrectly deferred samples) across all test samples.

**Results.** Figure 8a shows that as more uncertain predictions are deferred, the remaining predictions become more challenging, resulting in a decline in accuracy. This suggests that while the deferral strategy successfully excludes uncertain predictions, it also leaves a set of samples that are inherently harder to predict accurately. Additionally, Figure 8b demonstrates that as the uncertainty threshold increases, the true deferral rate rises, while the false deferral rate falls. This indicates that the model effectively identifies uncertain predictions (leading to more true deferrals) while reducing unnecessary deferrals. The point at which the true deferral rate surpasses the false deferral rate represents an optimal balance, maximizing decision quality and minimizing unwarranted deferrals. Combining the extended decision-making process under missing modality conditions (as presented in Appendix A.4.3), this analysis indicates that SURE's estimated uncertainty is a reliable indicator for ensuring high prediction quality.

# 7 Conclusion

**Contributions.** This work introduces SURE (Scalable Uncertainty and Reconstruction Estimation), which leverages pretrained multimodal frameworks for small datasets with missing modalities using latent space reconstruction. SURE integrates uncertainty estimation via a Pearson Correlation-based loss and error propagation, ensuring reliable predictions and adaptability across tasks and networks. It achieves state-of-the-art results in both downstream performance and uncertainty estimation.

**Limitations.** In developing SURE, we observed that certain modalities dominate the reconstruction process, making it easier to predict missing ones but causing significant performance drops when unavailable. This imbalance, unexplored in the current SURE framework, may limit the development of robust reconstruction modules and presents a valuable direction for future work.

## Impact Statements

By introducing uncertainty-aware multimodal learning, SURE has the potential to transform real-world AI deployment in high-stakes domains. Its ability to quantify uncertainty, propagate it through decision pipelines, and prevent overconfident mispredictions makes it an essential advancement for trustworthy AI in mission-critical applications. Future work can explore extending SURE to dynamic, real-time multimodal environments where uncertainty estimation plays a crucial role in adaptive decision-making.

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

# A  Appendix

## A.1  Related Works

**Multimodal missing modalities.** Recent research has focused on developing models resilient to missing modalities Ma et al. (2021; 2022); Poklukar et al. (2022); Woo et al. (2023); Lee et al. (2023); Li et al. (2024b); Lian et al. (2023); Li et al. (2024a). Key directions include: (1) Contrastive loss-based methods that align latent spaces for cross-modal knowledge transfer Ma et al. (2022); Lee & Van der Schaar (2021); Wang et al. (2020); (2) Generative approaches, such as VAE-based models Wu & Goodman (2018) or latent space reconstruction Woo et al. (2023), to approximate missing inputs; and (3) Prompt-based techniques Lee et al. (2023); Guo et al. (2024); Jang et al. (2024) that use trainable prompts to adapt models to various combinations of missing modalities. In the first direction, representative methods like Smil Ma et al. (2021) employ Bayesian meta-learning to approximate latent features for incomplete data, while GMC Poklukar et al. (2022) ensures geometric alignment in multimodal representations, allowing unimodal data to substitute for missing modalities. For the second direction, ActionMAE, inspired by the masked autoencoder framework Feichtenhofer et al. (2022); Bachmann et al. (2022), reconstructs latent representations of missing modalities by randomly dropping feature tokens and learning to predict them. In the third direction, Lee et al. Lee et al. (2023) propose missing-aware prompts that are integrated into pretrained multimodal transformers during training, enabling models to handle missing modalities effectively during evaluation. While these approaches show promise in specific scenarios, they often rely heavily on large-scale datasets and lack robust mechanisms for quantifying uncertainty in the presence of missing modalities. SURE addresses these gaps by leveraging pretrained models that require less data and providing a robust framework for estimating uncertainty in both reconstructed inputs and downstream predictions, enhancing reliability and interpretability.

**Uncertainty Estimation.** Recent methods for uncertainty estimation in predictions primarily rely on Bayesian models Lakshminarayanan et al. (2017); Kendall & Gal (2017c). However, while these models can estimate uncertainty, their predictive performance often lags behind other approaches. Some post-hoc works have explored using Laplace approximation to estimate uncertainty Daxberger et al. (2021); Eschenhagen et al. (2021), but these methods require computing the Hessian matrix, making them infeasible for high-dimensional problems Fu et al. (2018). Another direction involves test-time data augmentation Wang et al. (2019b); Ayhan & Berens (2018), where multiple outputs are perturbed to estimate uncertainty. However, this approach is sometimes poorly calibrated, which is critical for accurate uncertainty estimation Gawlikowski et al. (2023). SURE offers a more efficient alternative by estimating uncertainty without compromising predictive performance on downstream tasks. Unlike Laplace approximation, SURE avoids computational issues in high-dimensional spaces, and it does not rely on test-time perturbations, ensuring better-calibrated uncertainty estimates across diverse settings. Additionally, SURE imposes no assumptions on the output size, making it more flexible for a variety of applications.

## A.2  SURE's Additional Details

### A.2.1  Negative Log Likelihood Loss for Uncertainty Estimation

**Analysis for the convergence of $\mathcal{L}_{NLL}(.,.)$**

The detailed derivation of the gradient of $\mathcal{L}_{NLL}$ with respect to prediction $\tilde{y}_i$ is:

$$\frac{\partial \mathcal{L}_{NLL}(\tilde{y}, \tilde{\sigma}^2)}{\partial \tilde{y}_i} = \frac{\partial}{\partial \tilde{y}_i} \sum_{i=1}^{N} \frac{\tilde{\Delta}_i^2}{2\tilde{\sigma}_i^2} + \frac{\log\left(\tilde{\sigma}_i^2\right)}{2} = \frac{\tilde{\Delta}_i}{\tilde{\sigma}_i^2} = \frac{\tilde{y}_i - y_i}{\tilde{\sigma}_i^2}.$$

Solving $\frac{\partial \mathcal{L}_{NLL}(\tilde{y}, \tilde{\sigma}^2)}{\partial \tilde{y}_i} = 0$ give us the closed form solution $\tilde{y}_i^* = y_i$ (One can further verify sufficient condition $\frac{\partial^2 \mathcal{L}_{NLL}(\tilde{y}_i, \tilde{\sigma}_i^2)}{\partial \tilde{y}_i^2} = \frac{1}{\tilde{\sigma}_i^2} > 0$ hold true $\forall i$).

Similarly, gradient of $\mathcal{L}_{NLL}$ with respect to $\tilde{\sigma}_i^2$ is:

$$\frac{\partial \mathcal{L}_{NLL}(\tilde{y}, \tilde{\sigma}^2)}{\partial \tilde{\sigma}_i^2} = \frac{\partial}{\partial \tilde{\sigma}_i^2} \sum_{i=1}^{N} \frac{\tilde{\Delta}_i^{\,2}}{2\tilde{\sigma}_i^2} + \frac{\log\left(\tilde{\sigma}_i^2\right)}{2} = \frac{1}{2\left(\tilde{\sigma}_i^{\,2}\right)^2}\left(\tilde{\sigma}_i^2 - \tilde{\Delta}_i^2\right).$$

Setting $\frac{\partial \mathcal{L}_{NLL}(\tilde{y}, \tilde{\sigma}^2)}{\partial \tilde{\sigma}_i^2} = 0$ yield $\tilde{\sigma}_i^{2*} = \tilde{\Delta}_i^2$. Verifying the sufficient condition:

$$\left.\frac{\partial^2 \mathcal{L}_{NLL}}{\partial\left(\tilde{\sigma}_i^2\right)^2}\right|_{\tilde{\sigma}_i^2 = \tilde{\Delta}_i^2} = \frac{1}{2}\left(-\frac{1}{\left(\tilde{\Delta}_i^2\right)^2} + \frac{2\tilde{\Delta}_i^2}{\left(\tilde{\Delta}_i^2\right)^3}\right) = \frac{1}{2}\left(-\frac{1}{\tilde{\Delta}_i^4} + \frac{2}{\tilde{\Delta}_i^4}\right) = \frac{1}{2} \cdot \frac{1}{\tilde{\Delta}_i^4} > 0$$

This test result indicates a local minimum at $\tilde{\sigma}_i^{2*} = \tilde{\Delta}_i^2$.

The issue optimizing $\mathcal{L}_{NLL}(\tilde{y}, \tilde{\sigma}^2)$ come up when $\tilde{\Delta}_i \to 0$, this pull the gradient $\frac{\partial \mathcal{L}_{NLL}(\tilde{y}_i, \tilde{\sigma}_i^2)}{\partial \tilde{\sigma}_i^2}$ to the form $\frac{0}{0}$, which is mathematically undefined. This pose a significant issue for gradient-based optimization algorithms like Gradient Descent and cause arbitrary potential issues (gradient vanishing/exploding, numerical under/overflow sensitive to small changes of $\tilde{\Delta}^2$, etc).

**Analysis for the convergence of $\mathcal{L}_{PCC}(.,.)$**

For this analysis, we focus on the convergence for finding optimal $\tilde{\sigma}_i^{2*}$ of $\mathcal{L}_{PCC}(.,.)$. With:

$$\mathcal{L}_{PCC}(\tilde{\sigma}^2, \tilde{\Delta}^2) = 1 - r(\tilde{\sigma}^2, \tilde{\Delta}^2);$$

$$r(\tilde{\sigma}^2, \tilde{\Delta}^2) = \frac{\sum_{i=1}^{N}\left(\tilde{\sigma}_i^2 - \mu_{\sigma^2}\right)\left(\tilde{\Delta}_i^2 - \mu_{\Delta^2}\right)}{\sqrt{\sum_{i=1}^{N}\left(\tilde{\sigma}_i^2 - \mu_{\sigma^2}\right)^2}\sqrt{\sum_{i=1}^{N}\left(\tilde{\Delta}_i^2 - \mu_{\Delta^2}\right)^2}} := \frac{A}{B}.$$

We have:

$$\frac{\partial \mathcal{L}_{PCC}(\tilde{y}, \tilde{\sigma}^2)}{\partial \tilde{\sigma}_i^2} = -\frac{\partial r(\tilde{y}, \tilde{\sigma}^2)}{\partial \tilde{\sigma}_i^2} = -\frac{1}{B}\frac{\partial A}{\partial \tilde{\sigma}_i^2} + \frac{A}{B^2}\frac{\partial B}{\partial \tilde{\sigma}_i^2}.$$

$$\begin{aligned}
\frac{\partial A}{\partial \tilde{\sigma}_i^2} &= \frac{\partial}{\partial \tilde{\sigma}_i^2}\sum_{j=1}^{N}\left(\tilde{\sigma}_j^2 - \mu_{\sigma^2}\right)\left(\tilde{\Delta}_j^2 - \mu_{\Delta^2}\right) \\
&= \sum_{j=1}^{N}(\delta_{ij} - \frac{1}{N})(\tilde{\Delta}_j^2 - \mu_{\Delta^2}) \text{ (where } \delta_{ij} = 1 \text{ if } i = j \text{ else } 0) \\
&= \tilde{\Delta}_i^2 - \mu_{\Delta^2} \text{ (Since } \frac{1}{N}\sum_{j=1}^{N}\tilde{\Delta}_i^2 = \mu_{\Delta^2}).
\end{aligned}$$

Also,

$$\frac{\partial B}{\partial \tilde{\sigma}_i^2} = \frac{\partial}{\partial \tilde{\sigma}_i^2}\sqrt{\sum_{j=1}^{N}\left(\tilde{\sigma}_j^2 - \mu_{\sigma^2}\right)^2}\sqrt{\sum_{j=1}^{N}\left(\tilde{\Delta}_j^2 - \mu_{\Delta^2}\right)^2}$$

Denoting $\sigma_{\tilde{\sigma}^2} := \sum_{j=1}^{N} \left( \tilde{\sigma}_j^2 - \mu_{\sigma^2} \right)^2$, $\sigma_{\tilde{\Delta}^2} := \sum_{j=1}^{N} \left( \tilde{\Delta}_j^2 - \mu_{\Delta^2} \right)^2$, we have:

$$\frac{\partial B}{\partial \tilde{\sigma}_i^2} = \sigma_{\tilde{\Delta}^2} \frac{1}{2\sigma_{\tilde{\sigma}^2}} \frac{\partial}{\partial \tilde{\sigma}_i^2} \sum_{j=1}^{N} \left( \tilde{\sigma}_j^2 - \mu_{\sigma^2} \right)^2$$

$$= \sigma_{\tilde{\Delta}^2} \frac{1}{2\sigma_{\tilde{\sigma}^2}} \left[ \sum_{j=1}^{N} 2(\tilde{\sigma}_j^2 - \mu_{\sigma^2})(\delta_{ij} - \frac{1}{N}) \right]$$

$$= \sigma_{\tilde{\Delta}^2} \frac{1}{2\sigma_{\tilde{\sigma}^2}} \left[ 2(\tilde{\sigma}_i^2 - \mu_{\sigma^2}) - \frac{2}{N} \sum_{j=1}^{N} (\tilde{\sigma}_j^2 - \mu_{\sigma^2}) \right]$$

$$= \sigma_{\tilde{\Delta}^2} \sigma_{\tilde{\sigma}^2} \frac{\tilde{\sigma}_i^2 - \mu_{\sigma^2}}{\sigma_{\tilde{\sigma}^2}^2}$$

Assembling the results, we have:

$$\frac{\partial \mathcal{L}_{PCC}(\tilde{y}, \tilde{\sigma}^2)}{\partial \tilde{\sigma}_i^2} = -\frac{\partial r(\tilde{y}, \tilde{\sigma}^2)}{\partial \tilde{\sigma}_i^2} = -\frac{1}{B} \frac{\partial A}{\partial \tilde{\sigma}_i^2} + \frac{A}{B^2} \frac{\partial B}{\partial \tilde{\sigma}_i^2}$$

$$= -\frac{\tilde{\Delta}_i^2 - \mu_{\Delta^2}}{\sigma_{\tilde{\Delta}^2} \sigma_{\tilde{\sigma}^2}} + \frac{\tilde{\sigma}_i^2 - \mu_{\sigma^2}}{\sigma_{\tilde{\sigma}^2}^2} * \frac{\sum_{j=1}^{N} \left( \tilde{\sigma}_j^2 - \mu_{\sigma^2} \right) \left( \tilde{\Delta}_j^2 - \mu_{\Delta^2} \right)}{\sigma_{\tilde{\Delta}^2} \sigma_{\tilde{\sigma}^2}}$$

$$= -\frac{\tilde{\Delta}_i^2 - \mu_{\Delta^2}}{\sigma_{\tilde{\Delta}^2} \sigma_{\tilde{\sigma}^2}} + \frac{\tilde{\sigma}_i^2 - \mu_{\sigma^2}}{\sigma_{\tilde{\sigma}^2}^2} * r(\tilde{\sigma}^2, \tilde{\Delta}^2)$$

$$= \frac{1}{\sigma_{\tilde{\sigma}^2}} \left[ \frac{\tilde{\sigma}_i^2 - \mu_{\sigma^2}}{\sigma_{\tilde{\sigma}^2}} * r(\tilde{\sigma}^2, \tilde{\Delta}^2) - \frac{\tilde{\Delta}_i^2 - \mu_{\Delta^2}}{\sigma_{\tilde{\Delta}^2}} \right]$$

$$= \frac{1}{\sigma_{\tilde{\sigma}^2}} \left[ \sigma_{\tilde{\sigma}^2} * r(\tilde{\sigma}^2, \tilde{\Delta}^2) - \sigma_{\tilde{\Delta}^2} \right].$$

This last result suggest the gradient $\frac{\partial \mathcal{L}_{PCC}(\tilde{y}, \tilde{\sigma}^2)}{\partial \tilde{\sigma}_i^2}$ involves all standardized variables, which are within a manageable numerical range, reducing the risk of numerical instability. In addition, there is no divisions by $\tilde{\sigma}_i^2$, hence stabilize the training process even in the event when $\tilde{\Delta}_i^2 \to 0$.

### A.2.2 Reconstruction Modules.

SURE involves a set of reconstruction modules to best leverage the pretrained models' weights. Each reconstruction module is tailored for a specific modality, hence this reconstruction logic is linearly scale with the total number of modalities.

**Design.** While not mentioned in SURE logic, it should be noted that all $z^j$ are first linearly projected into a shared latent space wherever needed, before passing to the reconstruction modules. This step involves a single matrix multiplication done per modality, and the learnable matrix is trained together with the reconstruction modules. With that, all $r^i(.)$'s are working with the same input latent space, we unify the design of $r^i(.)$'s to be identical across different modalities. Specifically, the design of reconstruction module $r^i(.)$ is kept as simple as possible, with the major component as Fully Connected layers and ReLU activations as follow:

$$r_{share}^i(z^j) = \text{FC}(\text{ReLU}(\text{FC}(z^j))),$$
$$r_\mu^i(z^j) = \text{FC}(\text{ReLU}(\text{FC}(\text{ReLU}(r_{share}^i(z^j)))), \tag{14}$$
$$r_\sigma^i(z^j) = \text{SoftPlus}(\text{FC}(\text{ReLU}(\text{FC}(\text{ReLU}(r_{share}^i(z^j)||r_\mu^i(z^j))))).$$

In Equation 14, $||$ denotes the concatenation operation, and $SolfPlus()$ activation is used to ensure the positiveness of returned uncertainty.

**Complexity.** Below, we analyze the complexity of the chosen reconstruction modules. Table 11 lists hyperparameters involved in the analysis.

Table 11: $r^i(.)$ related hyper-parameters

| Notation | Description |
|---|---|
| $M$ | number of modalities |
| $L$ | number of FC layers (in total) |
| $d_i$ | hidden dimension of $i^th$ layer's output |
| $d_0$ | input dimension |

**Time Complexity.** Assume a single multiplication or summation operation can be performed in unit time ($\mathcal{O}(1)$). We have the calculation for number of operations in a forward pass as follows.

Within the $i^t h$ FC layer:
$$d_{i-1} * d_i + di,$$
Over $L$ layers:
$$\sum_{i=1}^{L} d_{i-1} * d_i + di.$$

In our implementations, we choose the same dimensions for all hidden outputs (same $d = d_i \forall i = 1, \ldots, L$), and there are $M$ modules $r^i(.)$. With this, the total number of operation is:

$$M \sum_{i=1}^{L} d_{i-1} * d_i + di = M * L * d * (d+1) = \mathcal{O}(M * L * d^2)$$

By utilizing matrix product and GPU acceleration, $d^2$ operations can in fact be performed in $\mathcal{O}(1)$ time, make the whole time complexity for individual branches be $\mathcal{O}(M * L)$, which is linearly scaled with $M$.

**Space Complexity.** Regarding the space complexity, within $i^{th}$ layer, beside the need for storing parameter matrix of size $(d_{i-1}+1) \times d_i$, output after performing $ReLU$ activation are also stored to later perform back-propagation. Hence, the total number of stored parameters is:

$$(d_{i-1} + 1) * d_i + d_i = (d_{i-1} + 2) * d_i.$$

Following similar derivation with $L$ layers and $M$ branches, replacing $d = d_i \forall i = 1, \ldots, L$, we have the total space complexity is:
$$M * L * (d + 2) * d = \mathcal{O}(M * L * d^2).$$

Despite utilizing straightforward reconstruction procedure, SURE demonstrates effective reconstruction in the latent space while maintaining an overall additional time and space complexity linearly scaled with $M$ - the number of all modalities and $L$ - the number of FC layers (6 in our implementation including both reconstruction and uncertainty heads).

### A.2.3 Extension to M modalities.

For extension to $M$ modalities, we train the reconstruction module using $\mathcal{L}_{rec}$. We use each of the available modalities as the ground-truth output and the rest available modalities as input to predict. In the second phase, we freeze all of the reconstruction modules and train the classifier head with $\mathcal{L}_{downstream}$. For each sample with missing modalities, we reconstruct them with remaining available ones, and perform simple average operation to obtain the final reconstruction. Algorithm 1 summarize the whole training process of SURE for $m \geq 2$ modalities.

### A.2.4 Additional Implementation Details

**SURE's Implementation Details.** In SURE, we reutilize pretrained multimodal frameworks chosen for specific tasks. The only replacement is the final layers producing prediction, since the classification task

---

**Algorithm 1** SURE training process

---

**Input**:
$\triangleright$ $\mathcal{D}_{train} = \{(\mathbf{x}_k^i); \mathbf{y}_k | i \in \mathcal{I}_k - \text{set of indices for available modalities in sample } k^{th}\}$.
$\triangleright$ $f^i(.)$ - frozen pretrained projectors; $r^i(.)$ - reconstruction modules ($i = 1, \ldots, M$).
$\triangleright$ $\omega(.)$ - frozen pretrained fusion module; $g(.)$ - classifier head.
**Output**:
$\triangleright$ $r^{i*}(.)$ - Trained reconstruction modules; $g^*(.)$ - Trained classifier head ($i = 1, \ldots, M$).

1: *Initialize $r^i(.)$'s and $g(.)$*
2: $\triangleright$ *Train reconstruction modules*
3: **for** mini-batch $\mathcal{B} \in \mathcal{D}_{train}$ **do**
4:     $l_{rec} \leftarrow 0$;
5:     **for** $i \in \{1, \ldots, M\}$ **do**
6:        $l_{rec}^i \leftarrow 0$;
7:        **for** $j \in \{1, \ldots, M\}; j \neq i$ **do**
8:           $\mathbf{z}_k^i = f^i(\mathbf{x}_k^i)$    $(\forall k : i \in \mathcal{I}_k)$;
9:           $\tilde{\mathbf{z}}_k^i, \tilde{\sigma}_k^i \leftarrow r^i(\mathbf{z}_k^j)$    $(\forall k : i, j \in \mathcal{I}_k)$;
10:          $l_{rec}^i \leftarrow l_{rec}^i + \mathcal{L}_{rec}(\mathbf{z}_i; \mathbf{z}_j)$;
11:        **end for**
12:        $l_{rec} \leftarrow l_{rec} + l_{rec}^i$;
13:     **end for**
14:     Backprop with $l_{rec}$;
15:     Optimizer step;
16: **end for**

17: $\triangleright$ *Freeze reconstructed modules $r^i(.)$;*
18: $\triangleright$ *Train classifier head*
19: **for** mini-batch $\mathcal{B} \in \mathcal{D}_{train}$ **do**
20:     $\mathbf{z}_k^i = f^i(\mathbf{x}_k^i)$    $(\forall i : i \in \mathcal{I}_k)$;
21:     For $\forall i, j; i \notin \mathcal{I}_k, j \in \mathcal{I}_k$:
22:        $\tilde{\mathbf{z}}_{j-k}^i, \tilde{\sigma}_{j-k}^i = r^i(\mathbf{x}_k^j)$;
23:        $\tilde{\mathbf{z}}_k^i = \texttt{average}(\tilde{\mathbf{z}}_{j-k}^i)$;
24:        $\tilde{\sigma}_{\tilde{z}_i}^2 = \texttt{average}(\tilde{\sigma}_{j-k}^i)$;
25:        $\tilde{\mathbf{y}}_k, \tilde{\sigma}_{\omega-k} \leftarrow g(\omega(\mathbf{z}_k^i, \tilde{\mathbf{z}}_k^j))$
26:     $\tilde{\sigma}_{input-k} \leftarrow \sum_{i \notin \mathcal{I}_k} \left(\frac{\partial \omega}{\partial \tilde{z}_k^i}\right)^2 \tilde{\sigma}_{\tilde{z}_i}^2$;
27:     $\tilde{\sigma}_{\tilde{y}_k}^2 \leftarrow \tilde{\sigma}_{input-k} + \tilde{\sigma}_{\omega-k}$;
28:     $l_{downstream} \leftarrow \mathcal{L}_{downstream}(\tilde{\mathbf{y}}_k; \mathbf{y}_k)$;
29:     $l_{y-pcc} \leftarrow \mathcal{L}_{PCC}(\tilde{\sigma}_{\tilde{y}_k}^2; l_{downstream})$;
30:     Backprop with $l_{y-pcc}$ and $l_{downstream}$;
        Optimizer step;
31: **end for**

---

might involve different number of classes, and there is an additional output head for estimation of output uncertainty.

**Sentiment Analysis.** This task involves predicting the polarity of input data (e.g., video, transcript). We use MMML Wu et al. (2024b) trained on the CMU-MOSEI dataset Zadeh et al. (2018) as the pretrained framework. SURE's reconstruction modules are added right after the projection modules - *Text/Audio feature networks* in original paper's language Wu et al. (2024b). Their fusion network are kept intact to leverage most pretrained weights as possible. We replace the last fully connected layer - classifier with two layers - one for the final output and one for estimated output uncertainty.

**Book genre classification.** This task involves classifying book genres based on their titles, summaries (text), and covers (images). We integrate SURE with MMBT Kiela et al. (2019), a pretrained framework on the MM-IMDB dataset Arevalo et al. (2020). MMBT is a bitransformer architecture, hence we consider the all the processing before positional embedding and segment embedding as the projection logic (refer to Kiela et al. (2019) for clearer architecture details), and add our reconstruction modules are inseted after this logic. The remaining transformer logic are considered fusion modules, and kept intact.

**Human Action Recognition.** This task involves identifying human actions based on recorded videos and sensor data. We use HAMLET framework Islam & Iqbal (2020), pretrained on the large-scale MMAct dataset Kong et al. (2019) for this task. HAMLET define their projection modules as *Unimodal Feature Encoders* Islam & Iqbal (2020). SURE's reconstruction modules are included right after these encoders, while retain their original MAT module.

**Implementation Details.** In our comparative evaluation, we incorporate several state-of-the-art approaches, each representing prominent strategies. The baselines are grouped into two categories, reflecting the key challenges addressed by SURE: (1) Reconstruction methods for missing modalities, and (2) Uncertainty estimation methods. The reconstruction techniques include ActionMAE Woo et al. (2023), DiCMoR Wang et al. (2023), and IMDer Wang et al. (2024). For uncertainty estimation, we evaluate against the Gaussian Maximum Likelihood method Kendall & Gal (2017b); Wang et al. (2019a), Monte Carlo Dropout

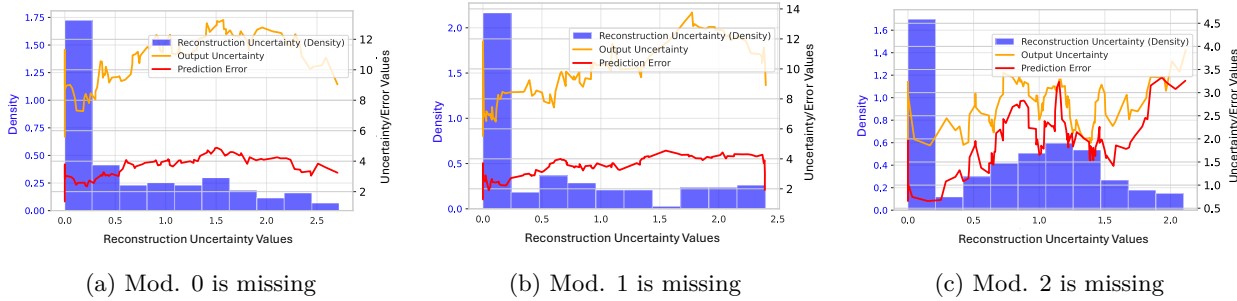

(a) Mod. 0 is missing        (b) Mod. 1 is missing        (c) Mod. 2 is missing

Figure 9: Inter-relationship between estimated output uncertainty, reconstruction uncertainty and output error on UTD-MHAD test dataset.

Maddox et al. (2019); Laves et al. (2019); Srivastava et al. (2014), and Ensemble Learning Lakshminarayanan et al. (2017). The original codebases of all baseline implementations are used for best reproducibility.

For all baselines, we also reutilize pretrained multimodal frameworks chosen for specific tasks like the adoptation with SURE. In addition, the hidden dimension used within Reconstruction-based baselines are also modified to be the same as those used within SURE for fair comparison.

**Hyperparameter Settings.** In all experiments, we randomly mask 50% of each modality during training. The same masked training sets are used consistently across SURE and all baselines to ensure fair comparison.

In SURE, the loss balancing coefficient $\lambda$–which controls the trade-off between uncertainty learning and task/reconstruction performance–is set to 0.5 across all tasks. This reflects the intuition that uncertainty estimation serves as a secondary objective relative to primary task performance. As shown in our empirical analysis (Appendix A.5), SURE remains stable and effective across a wide range of $\lambda \in [0.1, 0.7]$.

We also standardize the hidden dimension across methods per task: 1024 for sentiment analysis, 768 for book classification, and 2048 for human activity recognition. These dimensions are kept consistent across SURE and all compared baselines.

### A.3  Environment Settings

All implementations and experiments are performed on a single machine with the following hardware setup: a 6-core Intel Xeon CPU and two NVIDIA A100 GPUs for accelerated training.

Our codebase is primarily built using *PyTorch 2.0*, incorporating *Pytorch-AutoGrad* for deep learning model development and computations. We also use tools from *Scikit-learn*, *Pandas*, and *Matplotlib* to support various experimental functionalities. The original codebase for SURE will be released publicly upon publication.

### A.4  Additional Experiments and Analyses

#### A.4.1  Extended modalities missing scenarios

In Table 12, we provide a comprehensive evaluation of different frameworks across all combinations of input modalities on the UTD-MHAD dataset. This table expands on the information presented in Table 8 in the main text. The reported reconstruction uncertainty for cases with more than one available modality is averaged over all missing modalities (e.g., given (Video + Accel) inputs, the reported reconstruction uncertainty represents the average value for Gyro reconstruction). The results show that SURE consistently delivers the best performance in most uncertainty estimation scenarios while maintaining competitive results for the downstream task, underscoring its robustness across different missing modality situations.

Table 12: Results of different approaches on UTD-MHAD Dataset given every possible combination of input modalities.

| | Model | | F1↑ | Acc↑ | Reconstruct Uncertainty Corr↑ | Output Uncertainty Corr↑ |
|---|---|---|---|---|---|---|
| Modal Reconstruction | ActionMAE | Video | 0.044 | 0.059 | - | - |
| | | Accel | 0.204 | 0.231 | - | - |
| | | Gyro | 0.303 | 0.311 | - | - |
| | | Video + Accel | 0.034 | 0.085 | - | - |
| | | Video + Gyro | 0.301 | 0.305 | - | - |
| | | Accel + Gyro | 0.31 | 0.306 | - | - |
| | | Full | 0.531 | 0.537 | - | - |
| | DiCMoR | Video | 0.069 | 0.033 | - | - |
| | | Watch Accel | 0.473 | 0.408 | - | - |
| | | Phone Gyro | 0.52 | 0.472 | - | - |
| | | Video + Accel | 0.524 | 0.449 | - | - |
| | | Video + Gyro | 0.536 | 0.553 | - | - |
| | | Accel + Gyro | 0.577 | 0.586 | - | - |
| | | Full | 0.653 | 0.636 | - | - |
| | IMDer | Video | 0.089 | 0.069 | - | - |
| | | Watch Accel | 0.157 | 0.158 | - | - |
| | | Phone Gyro | 0.141 | 0.145 | - | - |
| | | Video + Accel | 0.152 | 0.152 | - | - |
| | | Video + Gyro | 0.248 | 0.257 | - | - |
| | | Accel + Gyro | 0.316 | 0.278 | - | - |
| | | Full | 0.687 | 0.689 | - | - |
| Uncertainty Estimation | SURE* + Gaussian MLE | Video | 0.116 | 0.074 | 0.166 | 0.122 |
| | | Watch Accel | 0.433 | 0.381 | 0.115 | 0.476 |
| | | Phone Gyro | 0.468 | 0.387 | 0.056 | 0.147 |
| | | Video + Accel | 0.432 | 0.443 | 0.104 | 0.237 |
| | | Video + Gyro | 0.462 | 0.502 | 0.095 | 0.143 |
| | | Accel + Gyro | 0.639 | 0.67 | 0.242 | 0.29 |
| | | Full | 0.693 | 0.651 | - | 0.292 |
| | SURE* + MC DropOut | Video | 0.156 | 0.09 | 0.122 | 0.136 |
| | | Watch Accel | 0.473 | 0.404 | 0.135 | 0.486 |
| | | Phone Gyro | 0.595 | 0.571 | 0.171 | 0.223 |
| | | Video + Accel | 0.452 | 0.52 | 0.186 | 0.292 |
| | | Video + Gyro | 0.546 | 0.56 | 0.101 | 0.376 |
| | | Accel + Gyro | 0.618 | 0.639 | 0.201 | 0.417 |
| | | Full | 0.739 | 0.718 | - | 0.512 |
| | SURE* + DeepEnsemble | Video | 0.25 | 0.207 | 0.249 | 0.126 |
| | | Watch Accel | 0.468 | 0.453 | 0.175 | 0.421 |
| | | Phone Gyro | 0.593 | 0.604 | 0.122 | 0.436 |
| | | Video + Accel | 0.652 | 0.662 | 0.092 | 0.346 |
| | | Video + Gyro | 0.776 | 0.781 | 0.176 | 0.462 |
| | | Accel + Gyro | **0.839** | **0.843** | 0.278 | 0.486 |
| | | Full | 0.737 | 0.735 | - | 0.481 |
| | **SURE** | Video | 0.161 | 0.121 | **0.878** | 0.226 |
| | | Watch Accel | 0.462 | 0.431 | 0.837 | 0.53 |
| | | Phone Gyro | 0.607 | 0.59 | 0.863 | 0.306 |
| | | Video + Accel | 0.542 | 0.606 | 0.873 | 0.412 |
| | | Video + Gyro | 0.609 | 0.637 | 0.862 | 0.379 |
| | | Accel + Gyro | 0.679 | 0.706 | 0.455 | 0.51 |
| | | Full | 0.739 | 0.74 | - | **0.568** |

### A.4.2 Extend analysis for inter-relationship between estimated uncertainties and error.

We further analyze Figure 9 to explore the inter-relationship between prediction error, reconstruction uncertainty, and output uncertainty. This visualization includes all test samples from the UTD-MHAD dataset

under different modality-missing scenarios, where each modality is systematically excluded. The histogram represents reconstruction uncertainty, while two line plots illustrate output uncertainty and prediction error.

Ideally, the two line plots should display an increasing trend, indicating a positive correlation with reconstruction uncertainty. Using SURE, such a trend is partially observed when modalities 0 and 1 are missing; however, it is less evident when modality 2 is missing. When combined with downstream performance for different input modality combinations, this experiment reveals two key insights:

- The Importance of Strong Modalities: Modality 2 (Gyro) plays a critical role in both downstream task performance and reconstructing other modalities. This suggests that stronger modalities are more effective in compensating for or reconstructing missing inputs to solve particular downstream tasks.

- Correlations Between Quantities: Output uncertainty is strongly correlated with prediction error, whereas reconstruction uncertainty shows a weaker correlation. This result aligns with our design: output uncertainty is directly trained to reflect prediction error, while reconstruction uncertainty may not be the primary source of error–model limitations and other factors can also contribute significantly.

These observations highlight the nuanced dynamics between uncertainties and error quantities produced with SURE.

### A.4.3 Extend Decision Making Analysis

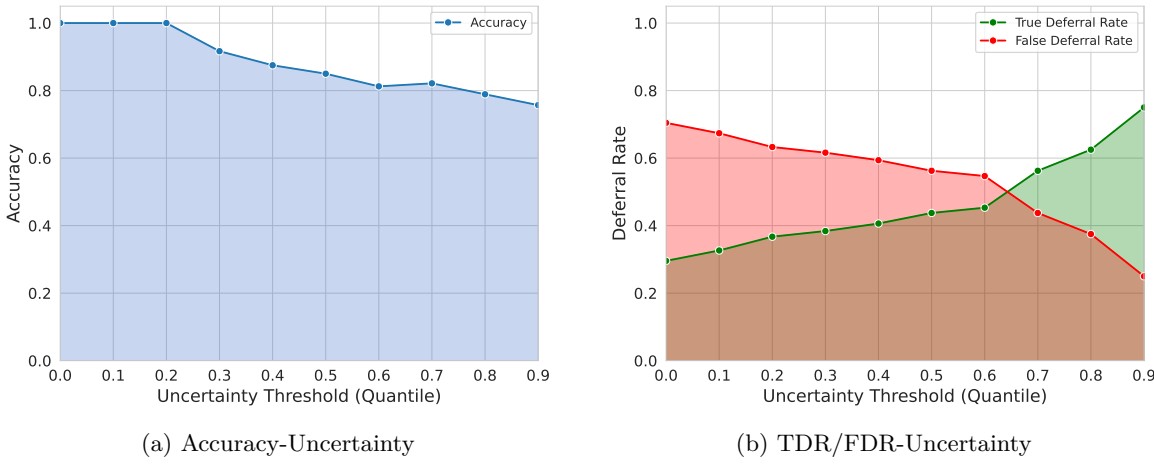

(a) Accuracy-Uncertainty (b) TDR/FDR-Uncertainty

Figure 10: Decision Making with uncertainty when Video is missing

Building on the main text analysis, we simulate the decision-making process on the UTD-MHAD dataset under conditions where different modalities are missing (Figures 10, 11, 12). Each figure represents the inference scenarios when the Video, Accel, or Gyro modality is absent. Similar to the decision-making process with full modalities, incorporating uncertainty estimates in cases with missing modalities continues to guide a reliable decision-making process by adjusting different uncertainty thresholds.

Lastly, for an aggregated perspective, the figure 13 shows the combined decision-making analysis on the UTD-MHAD dataset across different input modality combinations, with a deferral threshold set at the 0.65 quantile of output uncertainty. The green bars represent the true deferral rate, which indicates the proportion of incorrect predictions successfully deferred, while the red bars indicate the false deferral rate, representing the proportion of correct predictions unnecessarily deferred. The blue line shows the accuracy of the non-deferred predictions. From this visualization, the model effectively defers incorrect predictions when modality 0 is missing (true deferral rate $\sim 0.91$), but performance declines as more modalities are removed, with a notable drop for modality 2 (Gyro). False deferral rates remain low but vary slightly across scenarios at the

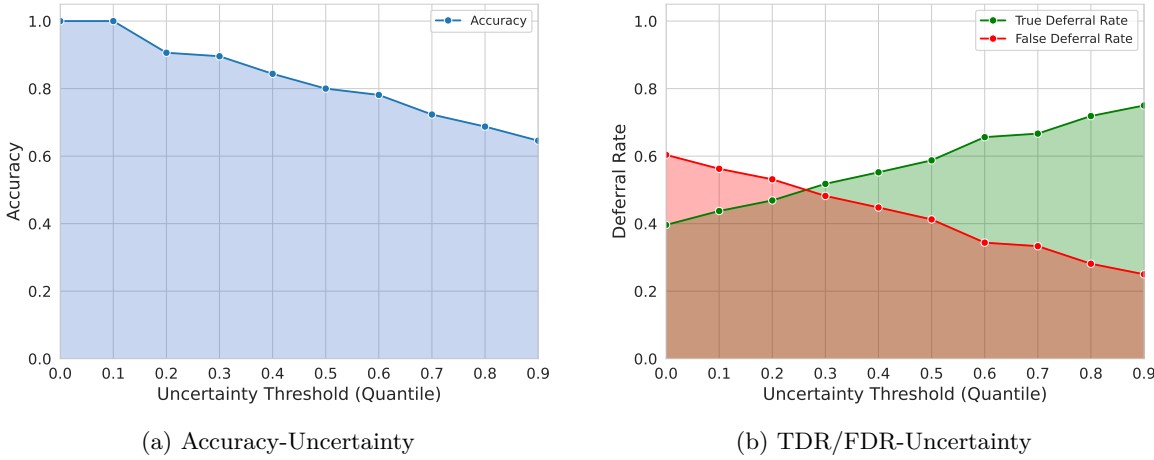

(a) Accuracy-Uncertainty

(b) TDR/FDR-Uncertainty

Figure 11: Decision Making with uncertainty when Accel is missing

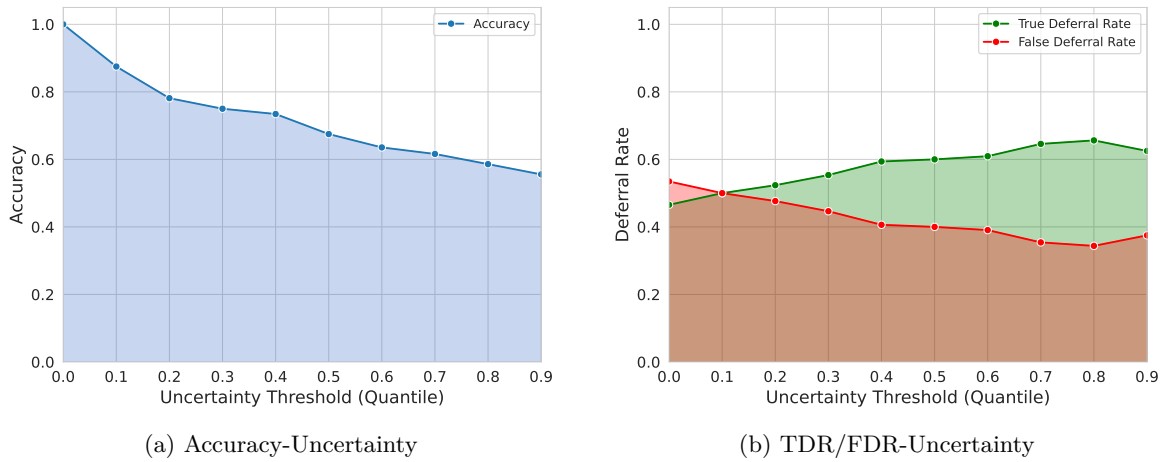

(a) Accuracy-Uncertainty

(b) TDR/FDR-Uncertainty

Figure 12: Decision Making with uncertainty when Gyro is missing

chosen threshold (0.65 uncertainty quantile), suggesting that the optimal uncertainty threshold may differ depending on the input modality combination. Non-deferred prediction accuracy decreases significantly when critical modalities like modality 2 are missing, underscoring its importance for robust performance. While the deferral strategy effectively reduces errors, further optimization of uncertainty thresholds is required to adapt to varying input modalities.

### A.5 Hyper-parameter Sensitivity Analysis

In Equations 9 and 10, the hyperparameter $\lambda$ controls the relative weight of uncertainty supervision compared to the reconstruction or downstream task loss. This section investigates how SURE's performance depends on the choice of $\lambda$, focusing on both downstream task accuracy and the quality of predicted uncertainty.

We conduct this sensitivity analysis on the CMU-MOSI dataset, varying $\lambda$ from 0.1 to 0.7. As shown in Table 13, increasing $\lambda$ slightly improves uncertainty calibration (e.g., PCC), while causing only marginal decreases in task performance. These results suggest that SURE is robust across a broad range of $\lambda$ values. Nevertheless, we recommend keeping $\lambda < 1$, as uncertainty estimation is intended to support–not dominate– the primary learning objective.

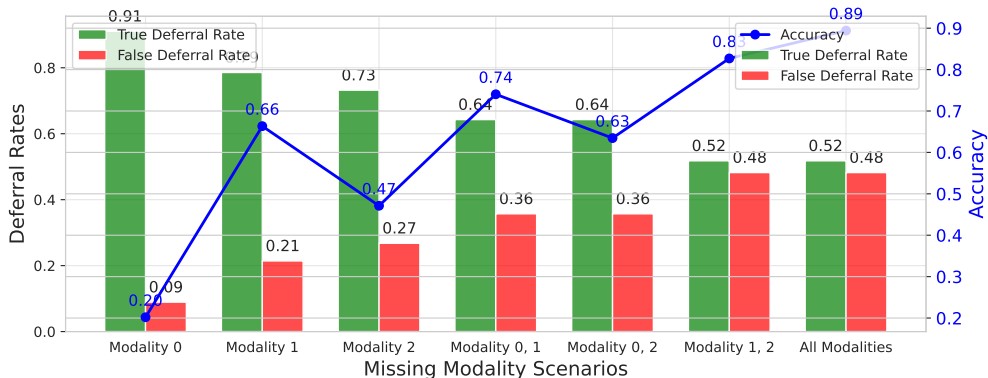

Figure 13: Decision making analysis with different input modalities combinations on UTD-MHAD dataset, with defer threshold set to be 0.65 quantile.

| $\lambda$ | MAE | | | Corr | | | F1 | | | Acc | | | R-Corr | | O-Corr | | | Output UCE | | |
|---|---|---|---|---|---|---|---|---|---|---|---|---|---|---|---|---|---|---|---|---|
| | T(ext) | A(udio) | F(ull) | T | A | F | T | A | F | T | A | F | T | A | T | A | F | T | A | F |
| 0.1 | **0.571** | **1.103** | **0.568** | **0.876** | 0.548 | **0.879** | 0.878 | 0.688 | 0.891 | 0.878 | 0.69 | 0.89 | 0.682 | 0.68 | 0.133 | 0.101 | 0.216 | 0.404 | 0.519 | 0.402 |
| 0.2 | 0.573 | 1.104 | 0.579 | 0.875 | 0.541 | 0.876 | 0.879 | **0.691** | **0.893** | 0.879 | **0.693** | **0.894** | 0.695 | 0.683 | 0.178 | 0.128 | 0.251 | 0.38 | 0.466 | 0.381 |
| 0.3 | 0.58 | 1.127 | 0.574 | 0.872 | 0.551 | 0.873 | 0.876 | 0.689 | 0.888 | 0.875 | 0.688 | 0.888 | 0.711 | 0.717 | 0.226 | 0.134 | 0.332 | 0.36 | 0.438 | 0.335 |
| 0.4 | 0.58 | 1.136 | 0.579 | 0.869 | 0.555 | 0.87 | 0.884 | 0.689 | 0.881 | 0.883 | 0.689 | 0.88 | 0.733 | 0.741 | 0.307 | 0.169 | 0.412 | 0.33 | 0.431 | 0.324 |
| 0.5 | 0.602 | 1.148 | 0.583 | 0.865 | 0.557 | 0.869 | **0.896** | 0.685 | 0.891 | **0.894** | 0.684 | 0.89 | 0.739 | 0.732 | 0.381 | 0.18 | 0.485 | 0.315 | 0.429 | 0.285 |
| 0.6 | 0.61 | 1.154 | 0.589 | 0.859 | **0.56** | 0.864 | 0.887 | 0.677 | 0.884 | 0.887 | 0.676 | 0.884 | **0.747** | 0.735 | 0.388 | 0.193 | 0.52 | 0.29 | 0.372 | 0.279 |
| 0.7 | 0.7 | 1.207 | 0.597 | 0.855 | 0.548 | 0.859 | 0.874 | 0.669 | 0.878 | 0.874 | 0.668 | 0.877 | 0.746 | **0.766** | **0.412** | **0.211** | **0.573** | **0.27** | **0.313** | **0.263** |

Table 13: Sensitivity analysis for hyper-parameter $\lambda$

### A.5.1 Additional Comparison with Prompt-based techniques

We further compare our SURE pipeline with two representative approaches that use prompt-based tuning techniques to address missing modalities Lee et al. (2023); Guo et al. (2024). Similar to our work, these approaches also leverage pretrained multimodal pipelines for efficient training. Their key innovation lies in introducing trainable prompts to indicate the presence of missing modalities.

**Setting.** The chosen task for demonstration is Semantic Analysis task. In line with the CMU-MOSI experiment described in the main text, both frameworks are implemented using the MMML Wu et al. (2024b) model, pretrained on the CMU-MOSEI dataset Zadeh et al. (2018). To ensure a fair comparison, all core modules from the original codebases of the two approaches are preserved to accurately replicate their performance. The training dataset is designed similarly to the main experiment, with 50% of modalities randomly masked and treated as missing.

**Result.** As shown in Table 14, SURE outperforms the two prompt-based approaches in handling missing modalities, achieving better downstream task performance. This advantage may stem from the limited number of learnable parameters introduced by these techniques, which likely constrain their ability to adapt effectively to scenarios with missing modalities.

Table 14: Additional results of different approaches on CMU-MOSI Dataset.

| Model | MAE↓ | | | Corr↑ | | | F1↑ | | | Acc↑ | | |
|---|---|---|---|---|---|---|---|---|---|---|---|---|
| | T(ext) | A(udio) | F(ull) | T | A | F | T | A | F | T | A | F |
| MPMM | 0.683 | 1.197 | 0.668 | 0.83 | 0.495 | 0.834 | 0.87 | 0.69 | 0.874 | 0.871 | 0.689 | 0.875 |
| MPLMM | 0.624 | 1.166 | 0.607 | 0.838 | 0.509 | 0.842 | 0.865 | **0.697** | 0.879 | 0.865 | **0.694** | 0.879 |
| **SURE** | **0.602** | **1.148** | **0.583** | **0.865** | **0.557** | **0.869** | **0.896** | 0.685 | **0.891** | **0.894** | 0.684 | **0.89** |

