# OpenReview forum: "SURE: Scalable Uncertainty Estimation for Multimodal Pretrained Pipelines with Missing Inputs"
_TMLR — Withdrawn by Authors_

### Review · Reviewer_1Xgt · 2025-08-10

**Summary Of Contributions:**

This work proposes a method (SURE) for adding support for missing modalities when fine-tuning a pre-trained multimodal model (assumed to have a number of independent modality processors and a final fusion stage) on data where some of those modalities are missing, and additionally for training this model to produce a *relative* confidence score indicating how much prediction error there is (relative to other samples).

The basic idea behind the approach is to insert "reconstruction modules" that use latent representations of known modalities to predict latent representations of missing modalities, and have these modules predict both an estimate of the representation as well as an estimate of the uncertainty of this prediction. Additionally, the authors add a learnable final output head that predicts both a best-guess output and a separate uncertainty term reflecting other aspects of the uncertainty not due to missing modalities. The authors propose training each of these uncertainty terms separately to be correlated with (but not identical to) squared error, then propagating the error from the reconstruction modules through the frozen pretrained model and output head using a linear approximation, then summing up the two uncertainty estimates.

**Audience:**

Yes

**Broader Impact Concerns:**

No broader impact concerns.

**Claims And Evidence:**

No

**Requested Changes:**

Critical changes:

- (for W1a) Adding two uncertainty estimates, where each uncertainty estimate was only trained using a Pearson-correlation-based loss, seems badly formed. As far as I understand it, you could scale any of these estimates by a constant without affecting the training loss, even though the sum of those constants might change significantly when used at evaluation time. This doesn't make sense to me as a design. If there is something else ensuring the scale of these uncertainty estimates is comparable, I think the authors should explain what it is; if there isn't, I think this means the method should be changed so that the sum is a meaningful quantity.
- (for W1b) The notion of "uncertainty" in this paper seems fairly fuzzy, and the authors state that they intentionally do not attempt to accurately estimate the actual variance, but the notation and proofs are written assuming these are real variance measurements. I think the authors should either ensure their approach is measuring an actual variance (and evaluate it accordingly), or avoid using variance notation like $\sigma^2$ for quantities that are not the variance of any random variable. Relatedly, Proposition 2.1 does not define what $\tilde{\sigma}^2_{input}$ actually is, and I believe this proposition is only correct if $\tilde{\sigma}^2_{input}$ was a real variance (even though the paper states that it is not intended to be a real variance).
- (for W1c) I think the authors should explain more clearly how their approach applies to classification problems, since this is one of the main application areas for the technique. The current draft explains things only as they relate to real-valued Gaussians, which behave very differently for uncertainty estimation.
- (for W2a) Please define the UCE metric and explain how the predicted variance is used when estimating calibration error (if at all).
- (for W2c and W2d) Please describe the other approaches being compared against in more detail. In particular, it would be important to explain the actual modifications to the SURE backbone in each case. Additionally, if uncertainty estimates are being compared across model variants that also make different predictions, it is important to include information about the predictive accuracy of those other variants and how the different uncertainty quantification approaches affect their overall performance.
- (for W3a) Please revise the introduction to more accurately reflect prior work.

Other changes that would strengthen the work:
- The first citation in the paper (Zong & Sun, 2023) has been retracted; I suggest using a different reference to motivate the paradigm.
- I am not convinced that Pearson correlation on its own is a good measurement of the quality of an uncertainty metric. Generally, one way to use uncertainty is to define a threshold and abstain from prediction if uncertainty is too high, but I don't think Pearson correlation is directly relevant to this. I think the results would be stronger if they used a more decision-relevant metric for uncertainty, for instance focusing more on the accuracy at a fixed selectivity level (e.g. how accurate is the method if it is allowed to defer X% of the time). The paper does discuss this somewhat in section 6 but it does not currently compare this between different uncertainty metrics.
- I am still fairly unsure how the uncertainty estimates produced by the proposed methods would enable better decision-making, or how an estimate of input-induced uncertainty would be used. I think the paper would be stronger if it gave more direct evidence for this.
- It is not clear to me what the graph in section 5.2 is showing. Were the PCC and NLL lines trained with the same objective or different objectives? Is this just showing that optimizing for a higher PCC leads to a higher PCC, whereas optimizing for a low NLL does not lead to a high PCC? If so, that seems fairly obvious (since optimizing any metric tends to make that metric increase), and I'm not sure it has implications about stability more broadly.
- It is strange to me that Section 5.2 refers to "a tendency toward overestimating both reconstruction and output uncertainties" given that the approach was designed only to give a relative measurement of uncertainty, not an absolute one?
- The toy problem is somewhat useful, but seems very simple, to the point where I am not sure it says much about the real-world properties of the system. I would find a more complex toy problem more convincing (e.g. training a small neural network on a low-dimensional dataset), although I admit this would make it impossible to have the closed-form oracle.

**Strengths And Weaknesses:**

**Strengths**

- **S1.** The proposed approach involves only adding a small number of additional learned parameters to an existing multimodal model, and makes it possible to use the multimodal model with missing modalities even when the original model assumed all modalities were present. This generally seems like a useful thing to be able to do.
- **S2.** The idea of propagating a learned error estimate through the frozen pretrained model via linearization is interesting; this seems like an inductive bias that could be fairly useful for estimating the total uncertainty due to missing modalities.
- **S3.** The authors include an application of their approach to a toy problem before going into the results on more realistic tasks, which helps build some intuition for what their model is doing.

**Weaknesses**

- **W1.** The proposed approach and loss function have some strange design choices that seem hard to motivate from an uncertainty-quantification perspective, and which do not seem like natural choices for the domains where the technique is being applied.
    - a) The authors motivate and explain their approach through the lens of "variance" (or squared error), and some of their derivations assume that uncertainty estimates are measuring the variance of a random variable. However, the authors also explicitly state that they do not intend their approach to actually estimate variance, and the loss they propose only encourages the uncertainty estimates to be *correlated* with variance/squared error. This makes the notation somewhat confusing, and Proposition 2.1 seems incorrect unless the method actually produces a meaningful variance estimate.
    - b) The approach also involves adding together multiple uncertainty estimates from different sources. This would make sense if the estimates were variance estimates, since variance is additive. However, in this case (as far as I can tell) the authors are not restricting the absolute magnitude of each uncertainty estimate at all, only their correlation, which makes the sum of these estimates not very intuitively meaningful.
    - c) The authors then apply their technique to classification problems. For a classification problem, it is somewhat unclear why "variance" should be a meaningful metric, especially given that the pointwise variance for binary decision variable is already uniquely determined by its expectation.
- **W2.** The evaluation criteria and baselines for the experiments are very unclear, and it is not possible to interpret the results based on the information provided.
    - a) The authors state that they use "Uncertainty–Calibration Error" (UCE) as a metric. However, this metric is not defined in the paper. The second time UCE appears, the authors cite Guo et al. (2017), but that cited work defines multiple metrics (ECE, MCE), and I did not find a definition of "UCE" in that work. Furthermore, the ECE and MCE metrics are specific to classification problems, and involve binning predicted probabilities and comparing predicted and actual expected values within those bins. This is not a procedure that can be applied directly to a variance-based metric, so it is not clear what value the authors are reporting here; these metrics also involve hyperparameters that were not discussed.
    - b) The authors additionally make multiple claims about the Pearson correlation between their uncertainty estimate and the true squared error. However, it is not clear to me that this is a meaningful metric for uncertainty estimation, or whether having a higher correlation would actually translate to being able to make better decisions based on the uncertainty estimate.
    - c) In section 4.1 the authors state that they compare against uncertainty estimates such as Gaussian Maximum Likelihood, Monte Carlo Dropout, and Deep Ensembles. They then present results comparing these methods to SURE by measuring UCE and Pearson correlation. However, they do not provide enough detail on what this means to understand what these results show. One particular question I have: are these other uncertainty estimates used to estimate the uncertainty of the SURE model, or do they involve training a separate model? If they train a separate model, it does not seem sensible to compare uncertainty quantification results without also comparing the actual predictions being made, since uncertainty is always a measurement relative to the prediction made by the model.
    - d) Similarly in section 4.1 the authors state that they compare to a number of reconstruction-based methods such as ActionMAE, DiCMoR, and IMDer, but also state that "all baseline methods are integrated into the same pretrained architectures used by SURE". There are very few details on what this integration means, so it is difficult to understand what this comparison is showing.
- **W3.** Some aspects of the motivation for the approach are badly supported or seem incomplete.
    - a) Much of the introduction revolves around a claim that missing inputs are a critical blocker for using pretrained multimodal models for real applications, but there is very little concrete evidence for this. Additionally, the authors claim that "a common strategy to handle missing modalities is to reconstruct them from the available ones, allowing the pretrained pipeline to operate as if the input were complete", and cite three previous papers as examples. However, from looking at these papers, none of them appear to actually use this strategy: Lian et al. (2023) and Woo et al. (2023) train models from scratch with joint reconstruction and prediction objectives (rather than reconstructing for the purpose of a pretrained pipeline), and Lee et al. (2023) fine-tune prompts that supplement/augment the missing modalities without trying to reconstruct them.
    - b) The authors claim that separating "input-induced uncertainty" from "model mismatch uncertainty" supports better decision making, but it is not clear to me in which domains this would be an important distinction to make or how this would be used, and the paper does not explain this further.

---

### Review · Reviewer_KipX · 2025-08-20

**Summary Of Contributions:**

The current manuscript introduces a new uncertainty estimation plug-in module, namely "SURE (Scalable Uncertainty
and Reconstruction Estimation) which i) reconstructs missing modalities in latent space and ii) produces prediction uncertainty decomposed into input-induced (from reconstruction) and model-mismatch (frozen backbone bias) components. Motivated by previous “fill-in the missing modality” methods don’t propagate reconstruction uncertainty to the final prediction and can deliver overconfident errors.
To this end, the SURE has two training phases: in phase 1, they train the reconstruction head with Eq.9, which in phase 2, freeze them and train the classifier head with task losses + $\lambda L_{PCC}$  (Eq. 10). Then, the method is evaluated on different tasks (sentiment analysis, classification and action recognition) and compared with diffrent UQ methods including (Gaussian-MLE/MC-Dropout/Deep Ensembles) with differet metrics (MAE, F1, ACC, Output-UCE).

**Audience:**

Yes

**Claims And Evidence:**

Yes

**Requested Changes:**

Please consider these points:
- Can you please evaluate the proposed method under structured missingness (blockwise/semantic), corruptions, and distribution shift (train/test domain gap) to probe model-mismatch uncertainty?
- Can you please provide risk–coverage curves, expected calibration error, alongside UCE to present coverage-aware trade-offs across thresholds and missingness patterns?
- Can you replace eplace the single-source mapper $r_j (.)$ with attention over all available latents, implemented as a mixture-of-experts; evaluate baseline averaging (current default) versus learned fusion?
- Please provide justification [or experimental results on e.g. CLIP] on the **scalability** of your method that can support this claim.

**Strengths And Weaknesses:**

Strengths :
- They address important and interesting problems.
- The paper is structured and written well.
- The proposed method meets minimal architectural intrusion: frozen backbone, small heads, latent-space operation; time/space overhead scales ~linearly with modalities.

Weaknesses and Limitations:
- Since SURE propagates uncertainty through a first-order Taylor approximation and treats reconstruction errors as independent and small, accuracy and calibration can suffer under nonlinear fusion or cross-modal error correlation!
- There is no empirical analysis for distribution shift, OOD detection, or heavy corruption beyond missingness!
- The PCC objective aligns rank-ordering of uncertainty with error but does not calibrate absolute variance; scale may be uncertain across datasets!
- Another weakness mentioned by authors is modality dominance, where performance can hinge on a “strong” modality; when it’s missing, both reconstruction and accuracy drop sharply
- The gradient or Jacobian estimate $\(\|\nabla_{\tilde z_j}\Omega\|^2\)$ can be noisy, with limited analysis of numerical stability or runtime overhead

-

---

### Review · Reviewer_NQpR · 2025-10-09

**Summary Of Contributions:**

This article introduces a scalable uncertainty estimation method SURE for multi-modality pretrained models when the inputs of a model have missing modality or there is input distribution mismatch. SURE is a lightweight and plug-and[play module which can enhance pretrained multimodal pipelines by predicting uncertainty from two sources: missing input uncertainty and distribution shift input uncertainty. The proposed methodology is evaluated on both toy and practical tasks, showing improved prediction accuracy and uncertainty calibration.

**Audience:**

Yes

**Claims And Evidence:**

No

**Requested Changes:**

See above.

**Strengths And Weaknesses:**

While the problem is well motivated, and it is quite easy to follow the main line of the article. Some technical points are not so clear, and it makes me hard to understand the proposed method, as well as the results.

### Lack of technical clarity:

- It is mentioned in the introduction that at the core of SURE, there is a shared latent space of the pretrained pipeline. By reading Fig 2, it is not so clear what this latent space is. Is it the output z^1 and z^2? In this case, the function f^1 and f^2 are not the same, what is shared between them?
- The motivation to replace the negative log-likelihood objective in eq 2 from the aspect of numerical instability at low error is not so clear. If one could solve this problem analytically as given by eq 3, why one needs to perform gradient-descent in this case? This motivation should be clarified.
- How is the model mismatch uncertainty is estimated by SURE? The phrase about this after eq. 4 is too quick and no detail is given in the rest of the article. Conceptually speaking, if the problem is classification, what would be the tilde y?
- The Pearson correlation based loss in eq. 6, what is the optimal solution? Should it be tilde sigma^2 = Delta^2? It seems that a constant factor times tilde sigma^2 could give the same optimal solution. Does that make any sense?
- Please clarify the choice of r_2(x_1) in section 3.2.
- The axe labels of Fig 4 are not so meaningful. It is hard to understand what they are from words.
- In Section 4.1, what are the differences between the baseline method Ouput COrr and R-Corr? Why a larger R-Corr indicates a better solution in Table 5? What is R-Corr?
- How is r chosen in Section 4.2?
### Confusion mathematical notation:
-	In Section 2.4, tilde(y) is used as a random variable, as well as a mean of the Gaussian conditional p(tilde y | bar x). This is quite confusing.
-	The Omega in Proposition 2.1 is not clear. It would be better to make it more precise.
-	There is a lack of summation over j in eq. 9. It is not s clear why there no such summation in eq. 10?
-	The introduced Perason correlation-based loss in eq. 7 lacks clarify. What is the tilde sigma_i^2 in relation to x and y?
### Insufficient support:
- In the abstract, tt is argued that SURE can deal with model mismatch uncertainty when there is distribution mismatch / insufficient adaption. This point is not fully clear as the distribution mismatch problem is not discussed in detail in the article. To include some evaluation on this point seems to be necessary.

- To make the motivation to replace the negative log-likelihood objective in eq 2 from the aspect of numerical instability at low error clear, it would be better to add some numerical results when Delta^2 -> 0 to show the stability of the proposed approach.

---

### Note · Authors · 2025-10-13

**Comment:**

Dear Editors and Reviewers,

Thank you for spending your precious time with our submission!

We really appreciate your invaluable comments and suggestions for our current manuscript.
Considering the amount of work we would take into account, we decide to withdraw our manuscript for a better, more detailed revision.

Best regards,
Authors.

**Withdrawal Confirmation:**

I have read and agree with the venue's withdrawal policy on behalf of myself and my co-authors.